# Carbon monoxide metabolism in freshwater anaerobic methanotrophic archaea

Reinier A. Egas[1], Heyu Lin[2], Andy O. Leu [2], Gene W. Tyson [2], Simon J. McIlroy [2] & Cornelia U. Welte [1]✉

Anaerobic methanotrophic archaea mitigate methane emissions in anoxic environments as key members of the biological methane filter. Despite their ecological significance, physiology of anaerobic methanotrophs remains poorly understood. Here, we demonstrate that the freshwater methanotroph 'Candidatus Methanoperedens BLZ2' prefers carbon monoxide (CO) over methane as an electron donor. Without respiratory nitrate, CO oxidation led to acetogenesis and methanogenesis with rates comparable to methane oxidation with nitrate. The circularized genome of 'Ca. M. BLZ2' encodes six Ni-dependent carbon monoxide dehydrogenases (CODHs), three of which were highly expressed. Furthermore, we identified a 156-kbp mobile genetic element carrying central metabolic gene clusters, including two additional, highly expressed CODHs. CODH genes were widespread in Methanoperedenaceae and showed diverse evolutionary affiliations, including Methanocomedenaceae anaerobic methanotrophs and bacterial lineages. These findings highlight CO metabolism and genome plasticity in anaerobic methanotrophs challenging their classification as obligate methanotrophs and their ecological role in anoxic carbon cycling.

Methane ($CH_4$) accounts for approximately 20% of global warming despite its low atmospheric concentration of 1936 ppb as of February 2025[1,2]. Understanding the microbial processes regulating $CH_4$ fluxes is critical for mitigating climate change. Anaerobic oxidation of $CH_4$ (AOM) is a key biogeochemical process that mitigates $CH_4$ emissions from anoxic environments by converting $CH_4$ to $CO_2$[3]. AOM provides a microbial methane filter that is estimated to reduce potential emissions by approximately 71% in anoxic marine sediments and up to 50% in freshwater wetlands[3,4].

AOM is mediated by a polyphyletic group of anaerobic methanotrophic archaea (ANME) within the phylum Halobacteriota, comprising Methanophagales (ANME-1), Methanocomedenaceae (ANME-2a/b), Methanogasteraceae (ANME-2c) in marine sediments, and Methanoperedenaceae (ANME-2d) in freshwater sediments[5–7]. Despite their ecological importance, current knowledge is largely confined to their basal functions in energy and central metabolism. This is primarily due to the lack of axenic cultures, necessitating reliance on molecular analyses of bioreactor or microcosm enrichments[7–11]. Among ANME, Methanoperedenaceae stand out for their metabolic versatility, coupling AOM to a range of terminal electron acceptors such as nitrate[11,12], metal(loid)s[9,10,13] and humic substances[14,15]. Unlike marine ANME, Methanoperedenaceae do not require a syntrophic partner and exhibit a pleomorphic life cycle[7]. Despite this respiratory flexibility, an overlooked aspect of ANME physiology is their potential use or preference for alternative electron donors. Recently, formate was proposed as an alternative donor coupled to nitrate reduction[16]. Intriguingly, methane oxidation was completely halted by formate addition. However, the enrichment also contained 'Ca. Methylomirabilis oxyfera' and 'Ca. Kuenenia stuttgartiensis', both of which encode formate dehydrogenases, with 'Ca. K. stuttgartiensis' being a potent formate oxidizer[17]. This underscores a knowledge gap in how alternative electron donors influence ANME metabolism and AOM activity

[1]Department of Microbiology, Radboud Institute for Biological and Environmental Sciences, Radboud University, Heyendaalseweg 135, Nijmegen, The Netherlands. [2]Centre for Microbiome Research, School of Biomedical Sciences, Queensland University of Technology (QUT), Translational Research Institute, Woolloongabba, QLD, Australia. ✉e-mail: c.welte@science.ru.nl

in anoxic environments. Currently, the presence of ANME archaea in molecular ecological studies is equated to a functioning anaerobic methane filter, yet studying factors that reduce the activity of this filter is still an emerging field[18].

ANME encode the Wood-Ljungdahl pathway (WLP) in addition to methyl-coenzyme M reductase that catalyzes the $CH_4$ activation reaction feeding carrier-bound methyl intermediates into the oxidative methyl branch of the WLP[5,19,20]. In bacteria, the WLP catalyzes carbon monoxide (CO) metabolism[21]. In *Clostridia*, a bifunctional Ni-dependent carbon monoxide dehydrogenase (CODH) reduces $CO_2$ to CO, which is then condensed by acetyl-CoA synthase (ACS) with a methyl group to form acetyl-CoA[22]. Bifunctional CODHs are encoded by *cdhA* in archaea or *acsA* in bacteria and form part of the CODH/ACS complex involved in both CO oxidation and acetyl-CoA synthesis[23,24]. Other anaerobes, such as *Carboxydothermus hydrogenoformans*[25] and *Deferribacter autotrophicus*[26], oxidize CO via monofunctional Ni-dependent CODHs, encoded by *cooS*, coupling CO oxidation to hydrogen production or respiration[27].

Within archaea, physiological evidence for acetogenesis remains limited, with an abundance of reports finding (near-)complete WLPs, supporting genomic predictions of acetogenesis in *Bathyarchaeota*, *Verstraetearchaeota,* and ANME[23,28,29]. Archaeal acetogenesis has been demonstrated in *Archaeoglobus fulgidus* during CO oxidation coupled to sulfate reduction[30] and in *Methanosarcina acetivorans* where at higher CO concentrations methanogenesis is inhibited[31]. In ANME-2a, transcriptomic and isotopic data from long-term incubations indicate trace amounts of acetate formation under AOM conditions[32]. We hypothesize that the presence of the WLP in ANME enables them to oxidize available CO, potentially impacting the efficiency of methane oxidation.

Here, we investigated whether freshwater anaerobic methanotrophic archaea utilize carbon monoxide as an electron donor under both respiratory and acetogenic conditions. We report on activity assays with biomass enriched in '*Ca*. Methanoperedens BLZ2',

metabolite analysis, metagenomics, and metatranscriptomics to provide a comprehensive analysis of carbon monoxide metabolism and prevalence in freshwater anaerobic methanotrophic archaea.

## Results

### '*Ca*. Methanoperedens BLZ2' oxidizes CO at high rates with respiratory and acetogenic metabolism

To assess substrate oxidation dynamics, batch activity assays were performed with granular biomass from an '*Ca*. M. BLZ2' enrichment culture (Fig. 1). Under nitrate respiratory conditions, the culture oxidized CO (14.3% vol/vol in the headspace) at high rates that reached 475 µmol d$^{-1}$ gDW$^{-1}$. Compared to that, the methane oxidation rate (in the absence of CO) proceeded more slowly at 210 µmol d$^{-1}$ gDW$^{-1}$ (Fig. 1A). When adding volumetric equal amounts of CO and $CH_4$ (7.1% vol/vol each) the CO oxidation rate (952 µmol d$^{-1}$ gDW$^{-1}$) far exceeded the $CH_4$ oxidation rate (43 µmol d$^{-1}$ gDW$^{-1}$) (Fig. 1B). This strong inhibition of $CH_4$ oxidation by CO mirrors previous observations of substrate preference in '*Ca*. M. nitroreducens' with formate[16]. To suppress bacterial activity, the following incubations were supplemented with 50 µg mL$^{-1}$ antibiotics (SVAK, streptomycin, vancomycin, ampicillin, kanamycin). During incubations with CO/$NO_3^-$ (CO oxidation rate 979 µmol d$^{-1}$ gDW$^{-1}$), no accumulation of $CH_4$, formate or acetate was detected (Fig. 1C). In incubations with $CH_4$/$NO_3^-$ ($CH_4$ oxidation rate 85 µmol d$^{-1}$ gDW$^{-1}$), $CO_2$ was the only oxidation product (Fig. 1C). In contrast, when '*Ca*. M. BLZ2' enriched biomass was incubated with CO in the absence of nitrate, supplemented with 50 µg mL$^{-1}$ antibiotics (SVAK), a pronounced shift in product formation was observed (Fig. 1D). CO oxidation rates (at 7.1% vol/vol) were even higher than under nitrate respiratory conditions and reached 1261 µmol d$^{-1}$ gDW$^{-1}$, accompanied by a rapid accumulation of formate during the first 20 hours, after which formate production proceeded at a much slower rate. Re-feeding of CO resulted in a gradual accumulation of formate. Intriguingly, $CH_4$ was produced at a rate in the same order of magnitude as methane oxidation would proceed (99 µmol d$^{-1}$ gDW$^{-1}$),

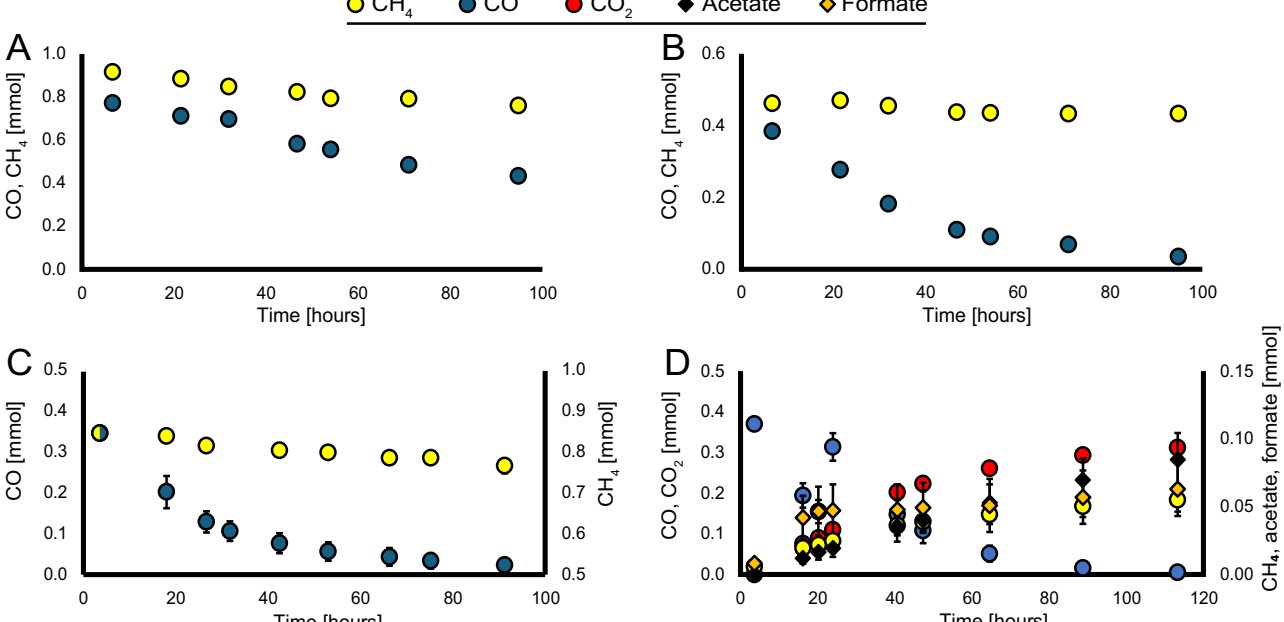

**Fig. 1 | Carbon monoxide oxidation by an enrichment culture of '*Ca*. M. BLZ2'.**
**A** CO or $CH_4$ oxidation in parallel batches with nitrate as electron acceptor ($n = 2$ biological replicates). Data are presented as mean values. **B** Both CO and $CH_4$, with nitrate as electron acceptor ($n = 2$ biological replicates); obtained in parallel to batch incubations for A. Data are presented as mean values. **C** CO or $CH_4$ with nitrate as electron acceptor and 50 µg mL$^{-1}$ antibiotics (SVAK) cocktail, ($n = 4$

biological replicates). Data are presented as mean ± standard deviation. During incubations with $NO_3^-$, no accumulation of $CH_4$, formate, or acetate was detected. **D** CO with $CO_2$ as electron acceptor and 50 µg mL$^{-1}$ antibiotics (SVAK) cocktail ($n = 3$ biological replicates); increase in $CO_2$ is depicted, after 20 h an additional 5 mL CO was added. Data are presented as mean ± standard deviation. Source data are provided as a Source Data file.

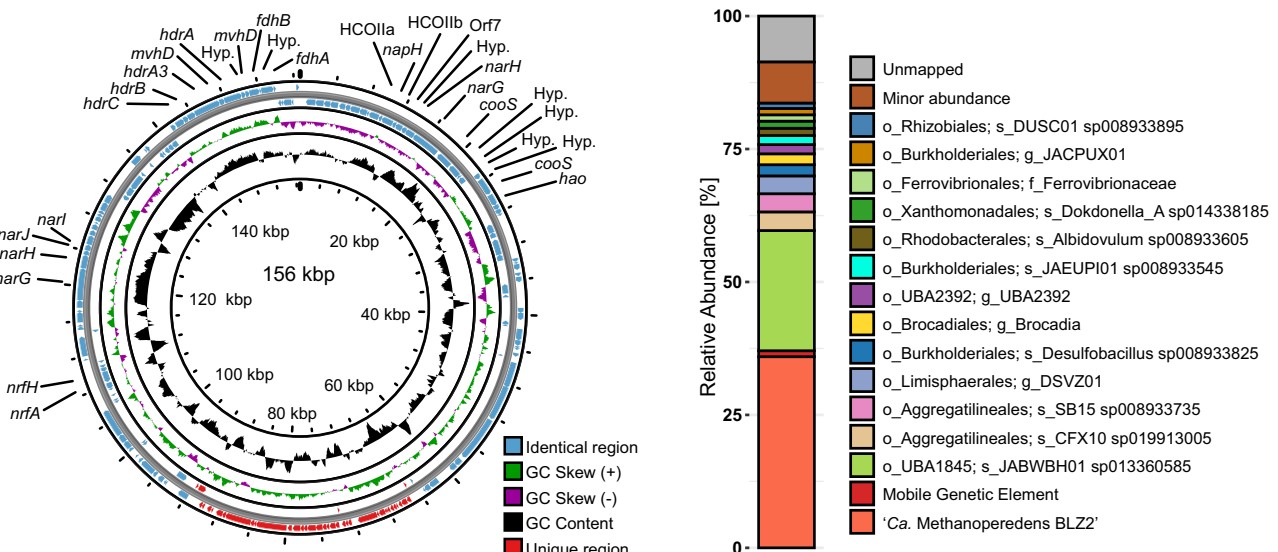

**Fig. 2 | Circularized mobile genetic element (MGE) associated with 'Ca. M. BLZ2'. Left:** Encoding multiple gene clusters with redox modules, including *cooS*, *narGHJI*, *nrfHA*, and a soluble heterodisulfide reductase cluster adjacent to *fdhAB*. Hyp. denotes genes encoding proteins of unknown function (hypothetical proteins), and *hao* indicates a hydroxylamine oxidoreductase-like multi-heme cytochrome. **Right:** DNA read-based relative abundance of the 'Ca. M. BLZ2' enrichment used as inoculum for activity assays. All MAGs with >1% relative abundance are shown; remaining MAGs are grouped as 'Minor abundance'. Source data are provided as a Source Data file.

exceeding the previously reported enzymatic back flux[33]. Additionally, acetate was produced with a near-linear rate throughout the incubation, indicating that in the presence of nitrate, CO-derived electrons are preferentially directed to respiratory (CO-coupled) nitrate reduction, whereas in nitrate-depleted conditions, CO oxidation leads to methanogenesis, acetogenesis, and formate overflow metabolism. The carbon balances during CO oxidation in the absence of nitrate were closed between consumed substrate and quantified products (Supplementary Data 4), excluding the formation of other metabolites.

### 'Ca. Methanoperedens BLZ2' encodes six carbon monoxide dehydrogenases in the genome

The metagenome-assembled genome of 'Ca. M. BLZ2' was circularized during this study. It contains a near-identical region to the previously described plasmid Hmp_v5[34], suggesting that this element either integrated into or was originally part of the 'Ca. M. BLZ2' genome. The genome contained five gene clusters encoding Ni-dependent carbon monoxide dehydrogenase (CODH) subunits (Supplementary Fig. 1). One gene cluster (*cdhABC* with *cdhDE* located elsewhere on the chromosome) encoded the bifunctional CO dehydrogenase/acetyl-CoA synthase enzyme integral to the Wood-Ljungdahl pathway for acetyl-CoA production. Four additional gene clusters were identified with genes encoding the monofunctional CooS; one of the clusters carried two copies of *cooS*, resulting in five gene clusters with six CODHs encoded in the genome of 'Ca. M. BLZ2'. The amino acid identity (AAI) between the CooS proteins ranged from 31.3-97.6%, indicating substantial sequence divergence (Supplementary Fig. 2). The cluster with two *cooS* copies also contained genes encoding a non-canonical nitrate reductase consisting of NarGH, NapH, membrane-bound heme copper oxidase subunits HCOIIa and b, as well as the archaeal Nar-specific Orf7; presumably, its active site is directed towards the extracellular space[35]. This colocalization suggests functional coupling of both enzymes. The two anaerobic CO-oxidizing model organisms *Carboxydothermus hydrogenoformans*[25] and *Methanosarcina acetivorans*[31] encode five CODHs in their genomes, with a similar high number of six CODHs in 'Ca. M. BLZ2'.

### A mobile genetic element associated with 'Ca. M. BLZ2' encodes multiple redox modules

Metagenome assembly revealed a 156-kbp circular contig, of which 129-kbp were near-identical to the 'Ca. M. BLZ2' chromosome (Fig. 2). Circularity was confirmed by Nanopore sequencing and PCR across the junction between genome-identical and unique regions. Distinct from other ANME-associated mobile genetic elements (MGEs), including Borgs (Supplementary Fig. 3)[36,37], this MGE encodes a unique combination of respiratory enzymes relevant for CO oxidation and nitrate reduction. Specifically, it includes two *cooS* genes as part of a cluster encoding the non-canonical nitrate reductase, a complete (canonical) nitrate reductase (*narGHJI*), a nitrite reductase (*nrfHA*), and a heterodisulfide reductase cluster (*hdrCBA3*, *mvhD*, *hdrA*, *mvhD*) adjacent to a formate dehydrogenase (*fdhAB*). All of these are in the shared region with the chromosome. An approximately 8-kbp region (positions 88-96 kbp) of the MGE contained multiple bacteriophage-related domains detected as prokaryotic viral orthologous groups (pVOGs), including two phage integrase family domains (Pfam: PF00589 and PF13495) (Supplementary Table 2). The unique region contains several short tandem repeats, which are smaller (20-33 bp) and show lower sequence identity than the long direct repeats (>200 bp) flanking previously described ANME-associated MGEs[34,38,39]. This MGE encodes versatile redox modules and could be a vector for augmenting respiratory metabolism via lateral gene transfer within the *Methanoperedenaceae*[40,41]. Metagenomics showed that 'Ca. M. BLZ2' was the dominant MAG with a relative abundance of 36% (Fig. 2). The associated MGE accounted for 1% of the total reads. The coverage across MGE and the genome was similar, and together with the sequence identity of most of the MGE suggests 'Ca. M. BLZ2' is its host.

### Increased relative transcriptional activity of 'Ca. M. BLZ2' under CO-oxidizing conditions

Transcriptomics showed an increase in the relative transcriptional activity of 'Ca. M. BLZ2' under CO-oxidizing conditions, from (average ± standard deviation) 36.4% ± 4.4% under $CH_4/NO_3^-$ to 56.1% ± 7.2% under $CO/NO_3^-$, and 58.6% ± 1.0% under $CO/CO_2$ (Fig. 3 and

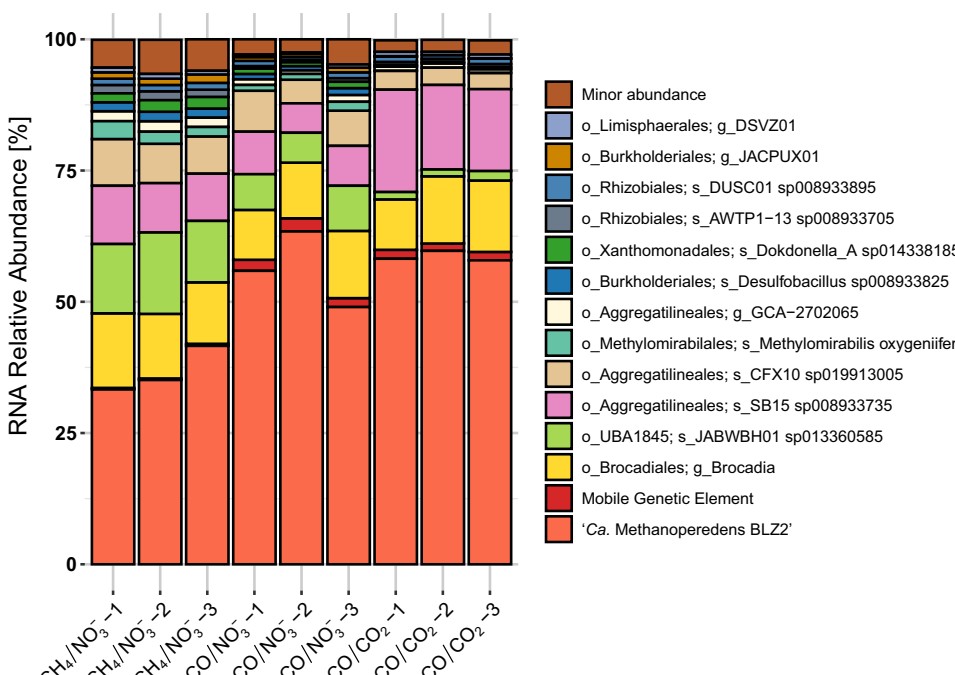

**Fig. 3 | Overall transcriptional activity of microbial community members under different conditions.** The relative abundance of mapped RNA reads to MAGs and the MGE is plotted. The top 10 most transcriptionally active MAGs per condition are shown across all conditions; remaining MAGs are grouped as 'Minor abundance'. Source data are provided as a Source Data file.

Supplementary Data 5). The MGE also increased in RNA read abundance from 0.3% under $CH_4/NO_3^-$ to approximately 2% under CO oxidizing conditions. Notably, '*Ca*. Methylomirabilis oxyfera' (GTDB annotation *Methylomirabilis oxygeniifera*) was transcriptionally active under $CH_4/NO_3^-$ (2% ± 0.8%) but dropped to 0% under $CO/CO_2$, making '*Ca*. M. BLZ2' the sole active methanotroph and archaeon under these conditions.

## Metatranscriptomic analysis of the bacterial community suggests a minor contribution to CO oxidation

To evaluate potential contributions to CO oxidation from the bacterial community, MAGs were screened for genes encoding aerobic Mo-CODHs and anaerobic Ni-CODHs. Mo-CODHs have a high affinity for CO[42] and can be active under nitrate respiratory conditions[43]. In total, 15 Mo-CODHs (*coxL*) were identified across 11 MAGs in our study, in addition to one non-ANME monofunctional Ni-CODH (*cooS*) (Supplementary Fig. 1 and Supplementary Data 6). Overall, these showed very low expression compared to the '*Ca*. M. BLZ2' Ni-CODH genes (Supplementary Data 6 and 7). The remaining three bifunctional Ni-CODHs were within complete CODH/ACS gene clusters in Chloroflexota MAGs (*CFX10* and *SB15*, order Aggregatilineales) and a Planctomycetota MAG (*Brocadia*, order Brocadiales). The *acsA* of *CFX10* and *SB15* was upregulated under $CO/NO_3^-$ and $CO/CO_2$ conditions. These were the most transcriptionally active community members after '*Ca*. M. BLZ2' (Fig. 3 and Supplementary Data 7). However, their CODH expression levels remained low compared to the high expression of '*Ca*. M. BLZ2'.

## Redox partitioning underlies CODH regulation and directs methane, acetate, and formate production

'*Ca*. M. BLZ2' encodes eight CODHs, including those on the MGE. Six, including *cdhA*, showed expression levels above 50 TPM under $CH_4/NO_3^-$ indicating constitutive expression (Fig. 4). Under $CO/NO_3^-$, TPM values strongly increased for five CooS-type CODHs. The upregulation of nitrate reductase gene clusters is in line with the observed increased respiratory activity rates with CO compared to $CH_4$ as substrate (Supplementary Data 8 and 9). The nitrite reductase gene *nrfA*, present

on both the chromosome and MGE, was downregulated. One of the pyruvate ferredoxin oxidoreductase genes (*porA*), a central anabolic entry point from acetyl-CoA, was upregulated under $CO/NO_3^-$ compared to $CH_4/NO_3^-$ and another one was upregulated under $CO/CO_2$ compared to $CH_4/NO_3^-$. Together, this indicates that '*Ca*. M. BLZ2' prioritizes respiratory electron transfer and carbon assimilation under $CO/NO_3^-$.

The transcriptome under $CO/CO_2$ conditions was profoundly different from the two respiratory conditions with many changes in central carbon metabolism, nitrogen metabolism, and redox homeostasis. Likely caused by switching from a nitrate-rich to a nitrate-depleted environment after prolonged enrichment on nitrate-driven AOM. CO is a known toxic gas and can inhibit growth in methanogens, where it can trigger detoxification responses facilitated by *cooS*. However, we did not observe a consistent or pronounced pattern change in expression of canonical stress, DNA and protein repair or cell division-associated genes under $CO/NO_3^-$ and $CO/CO_2$ conditions (Supplementary Data 9). While a subset of stress-associated genes was upregulated and expression of some cell-division genes was altered, this pattern is more consistent with a moderate physiological adjustment to CO exposure, or the nitrate-depleted redox environment, rather than a generalized stress response. Instead, transcriptional changes were dominated by redox and central metabolic pathways. Additionally, under $CO/CO_2$ conditions, the '*Ca*. M. BLZ2' *cooS* genes were downregulated but still expressed at similar TPM compared to the conditions with $CH_4/NO_3^-$ (Fig. 4). Genes encoding enzymes of the WLP were also similarly expressed, accompanied by a strong decrease in expression of the *mcr* genes encoding for the key methanotrophic enzyme methyl-CoM reductase (Supplementary Data 9). Nitrogen assimilation was also affected, with increased expression of glutamate dehydrogenase genes (*gdhA*) and downregulation of ammonium transporter (*amtA*) and nitrogenase genes (*nifH*). Carbon and redox flux were likely affected by the upregulation of genes encoding acetate ligase (*acdAB*), formate dehydrogenase (*fdhA*), and cytoplasmic heterodisulfide reductases (*hdrABC*). Under CO conditions, a phosphonate ABC transporter (*phnCDE*) colocalized with two high-affinity iron

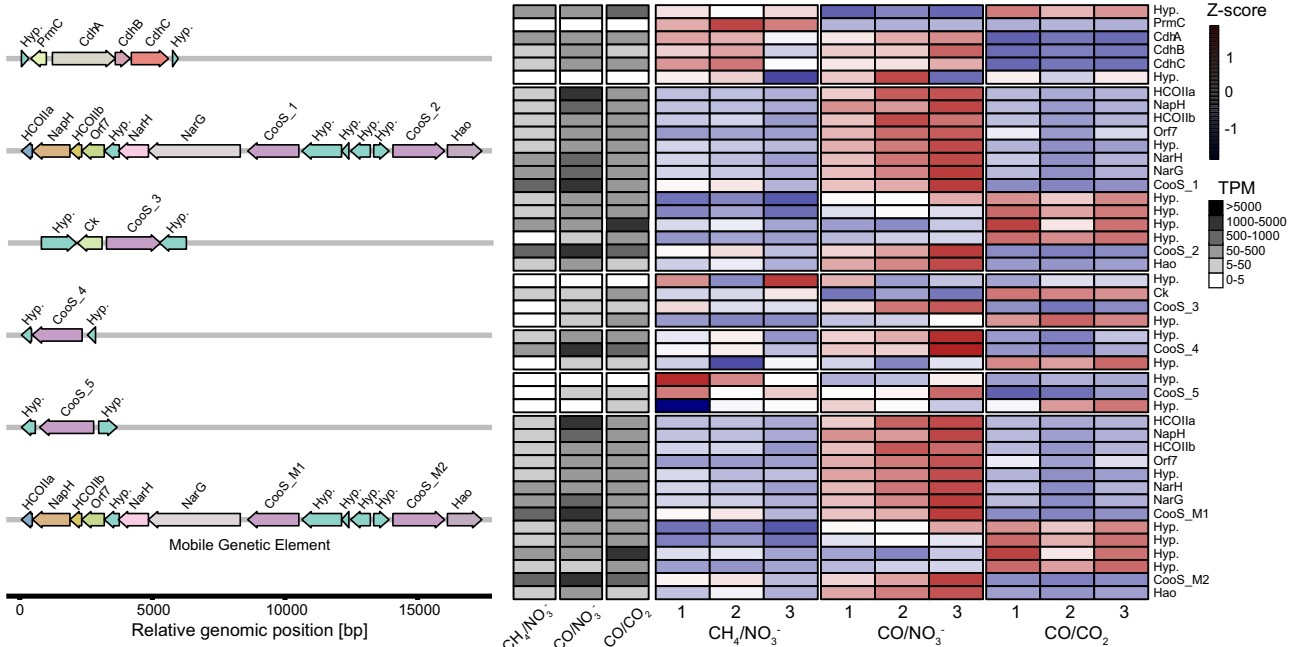

**Fig. 4 | Transcriptional response of *cooS* and *cdh* gene clusters in '*Ca*. M. BLZ2' under varying conditions. Left:** Overview of different gene clusters that encode Ni-CODH. The CODH/ACS *cdhABC* cluster is at the top, including the chromosomal Ni-CODHs; the bottom CODH and nitrate reductase cluster was identified on the MGE. Hyp. denotes genes encoding proteins of unknown function (hypothetical proteins), Hao a hydroxylamine oxidoreductase-like multi-heme cytochrome, Ck a carbamate kinase and PrmC a putative PrmC-like methyltransferase. Each gene cluster is normalized to relative genomic position for visualization purposes. **Right:** Heatmap with gene expression profiles of the different clusters per condition ($n = 3$ biological replicates). Detailed values are depicted in Supplementary Data 9. Heatmap color codes represent z-score-normalized per-row values; the average TPM per condition for each gene is depicted in grayscale. Source data are provided as a Source Data file.

permeases was among the most strongly induced, indicating that CO respiration drives concurrent iron and phosphorus demand and activates auxiliary nutrient-scavenging pathways.

## Metabolic model of fermentative carbon monoxide metabolism in '*Ca*. M. BLZ2'

Under $CO/CO_2$ conditions, '*Ca*. M. BLZ2' does not have access to external electron acceptors (except $CO_2$), therefore electrons derived from CO oxidation need to be redirected to form the fermentation products formate, acetate, and $CH_4$ (Fig. 5). In addition, energy needs to be conserved via alternative reactions than the membrane-bound electron transport chain. The primary energy-conserving reaction is acetate formation proceeding from acetyl-CoA via the ATP-dependent acetyl-Coenzyme A synthetase (*acs*) or acetate ligase (*acd*), releasing ATP. Methanogenesis, in addition, provides a $Na^+/H^+$ motive force through the activity of methyltransferase (*mtr*) and the membrane-bound heterodisulfide reductase (*hdrDE*), respectively, making acetate and methane fermentation products necessary for energy conservation. Formate production likely results from redox homeostasis reactions that prevent metabolic arrest. The oxidation of CO produces an excess of reduced redox carriers that are re-oxidized by enzymes in the WLP, leading to acetate and methane production. During methanogenesis, the CoM-S-S-CoB heterodisulfide is produced and must be reoxidized. This can be accomplished by one of the cytoplasmic electron bi- or confurcating heterodisulfide reductase complexes (HdrABC), producing formate, or by the membrane-bound heterodisulfide reductase (HdrDE). Additional energy could be conserved through membrane-bound oxidation of $F_{420}H_2$ released from ferredoxin re-oxidation reactions through electron bifurcation or the activity of potential cytoplasmic FqoF[44]. Formate can also be produced through cytoplasmic formate dehydrogenases. The acetogenic metabolism of freshwater anaerobic methanotrophs operates without the need for membrane-bound ferredoxin oxidation through the Rnf complex or Ech hydrogenase as required in acetogenic bacteria[45,46] and is therefore fundamentally different on a molecular level.

## Carbon monoxide dehydrogenases are widespread across freshwater anaerobic methanotrophs

The presence of Ni-CODH and nitrate reductase encoding genes were assessed in 36 publicly available *Methanoperedenaceae* genomes (Fig. 6). Parts of the bifunctional CODH/ACS complex, encoded by *cdhABCDE*, were conserved in all genomes, with most encoding all subunits, underscoring WLP as core metabolic trait. Apparent gene absences may reflect low completeness or high fragmentation since CODH/ACS is required for carbon assimilation from $CO_2$ and $CH_4$[47]. Monofunctional CODHs (*cooS*) were widespread and detected, from single to five copies, in 29 of 36 genomes (81%); the ones lacking *cooS* were highly fragmented, making true absence uncertain. There was no consistent link between either nitrate reductase types or the number of *cooS*. While the *cooS/narG* gene cluster suggests CO oxidation coupled to nitrate respiration the prevalence of *cooS* indicates a wider metabolic potential for CO oxidation in *Methanoperedenaceae* beyond nitrate respiration, including acetogenesis, CO-dependent metal(loid) reduction or CO-dependent direct interspecies electron transfer to syntrophic partners. Phylogenetic classification of the '*Ca*. M. BLZ2' CODHs showed different evolutionary origins, including *Methanocomedenaceae* and bacterial lineages (Supplementary Data 10). Multiple *Methanocomedenaceae* encode close homologs to the highly expressed CODHs in this study, demonstrating CODH prevalence and likely functionality in marine anaerobic methanotrophs, expanding the importance of CO metabolism across ANME. Both the distribution of *cooS* and its variable genomic contexts suggest that lateral gene transfer plays a role in CODH diversity across *Methanoperedenaceae*. This observation aligns with earlier findings that *Methanoperedenaceae* have an extensive genomic and metabolic plasticity not found in other anaerobic methanotroph clades[5,40]. The ability of

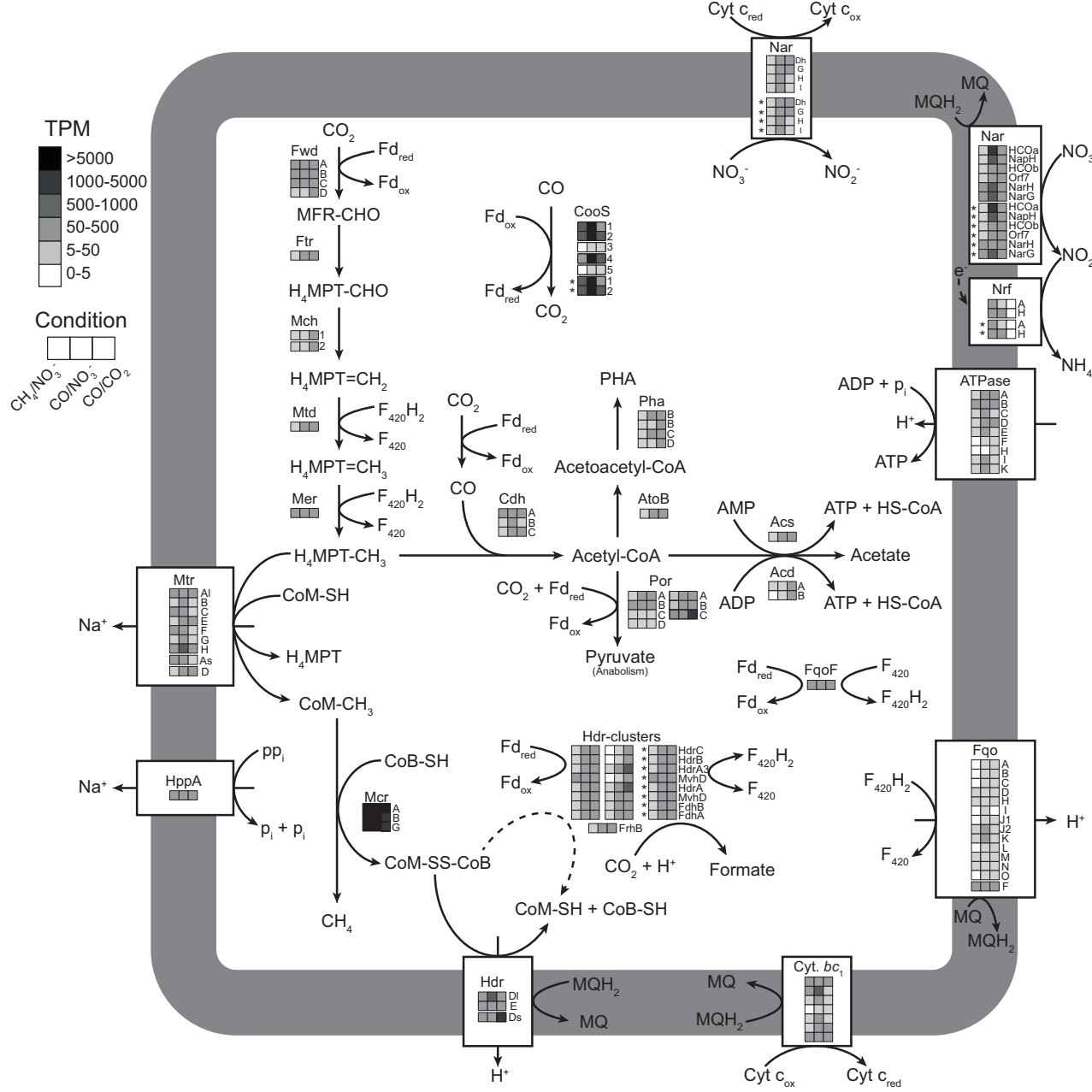

**Fig. 5 | Hypothesized metabolic model of CO metabolism of 'Ca. M. BLZ2'.** Boxes grouped together indicate genes co-localized; the asterisk indicates genes encoded on the MGE. The average TPM per condition for each gene is depicted by grayscale. Abbreviations are as follows: CooS, monofunctional carbon monoxide dehydrogenase; Fwd, formylmethanofuran dehydrogenase; Ftr, formylmethanofuran-$H_4$MPT formyltransferase; Mch, methenyl-$H_4$MPT cyclohydrolase; Mtd, $F_{420}H_2$-dependent methylene-$H_4$MPT dehydrogenase; Mer, methylene-$H_4$MPT reductase; Mtr, membrane-bound methyltransferase; Mcr, methyl-CoM reductase; HppA, pyrophosphatase; Cdh, acetyl-CoA synthase; PHA, polyhydroxyalkanoate; AtoB, acetyl-CoA acetyltransferase; PhaB, acetoacetyl-CoA reductase; PhaCE, polyhydroxyalkanoate synthase; Por, pyruvate: ferredoxin oxidoreductase; Acs, acetyl-coenzyme A synthetase; Acd, acetate ligase; Hdr, heterodisulfide reductase; FrhB, $F_{420}$-reducing hydrogenase; Mvh, methylviologen-reducing hydrogenase; Fdh, formate dehydrogenase; Fqo: $F_{420}$:quinone oxidoreductase; Cyt. $bc_1$, Rieske cytochrome $bc_1$ complex; Nar, nitrate reductase; Nrf, nitrite reductase; $Fd_{ox}/Fd_{red}$, ferredoxin oxidized/reduced; MFR, methanofuran; $H_4$MPT, tetra-hydromethanopterin; CoB-SH, coenzyme B; CoM-SH, coenzyme M; CoM-S-S-CoB, heterodisulfide; Cyt $c_{ox}$/Cyt $c_{red}$, Cytochrome c oxidized/reduced; MQ/$MQH_2$, menaquinone/menaquinol. Detailed names and expression levels are depicted in Supplementary Data 9. Source data are provided as a Source Data file.

*Methanoperedenaceae* to oxidize CO and $CH_4$, alongside the broader use of terminal electron acceptors across a wide redox range, likely contributes to their success in diverse anoxic environments.

### Etymology
Me.tha.no.per.e'dens. N.L. pref. *methano-*, pertaining to methane; L. pres. part. *peredens*, devouring; N.L. masc. n. *Methanoperedens*, a methane-devouring organism. car.bo.xy.di.vo'rans. N.L. neut. n. *carboxydum*, carbon monoxide; L. pres. part. *vorans*, devouring, digesting; N.L. masc. part. adj. *carboxydivorans*, digesting carbon monoxide.

### Discussion
In this study, we discovered that CO is a potent inhibitor of AOM for 'Ca. Methanoperedens BLZ2', henceforth called 'Ca. M. carboxydivorans', as it is preferentially used when $CH_4$ and CO are present. The

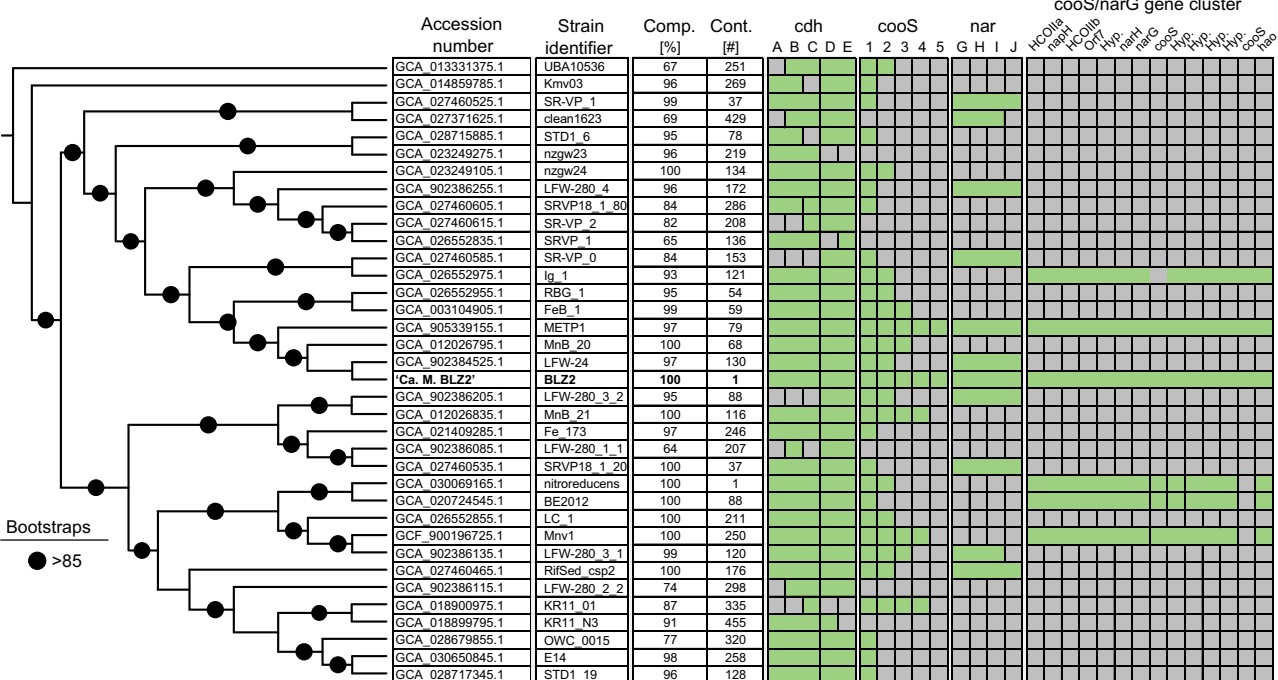

**Fig. 6 | Genome comparison of publicly available *Methanoperedenaceae* genomes and obtained genome in this study indicating widespread distribution of Ni-CODHs.** The phylogenetic tree is mid-point rooted, with branch lengths omitted for visualization. Bootstrap values represent support percentages from 1000 replicates. For each genome, estimated genome completeness (%) and contig count are indicated, with '*Ca*. M. BLZ2' depicted in bold. Green boxes indicate gene presence; merged boxes specify genes co-localized in a cluster. Gray boxes show gene absence: *cdhABCDE*, archaeal bifunctional CODH/ACS; *cooS*, monofunctional CODH; *narGHIJ*, nitrate reductase. The presence of a *cooS*/*narG* cluster is indicated separately and does not affect the number of *cooS* genes encoded. Apparent gene absences in some genomes may reflect low completeness or high fragmentation.

genome of '*Ca*. M. carboxydivorans is ideally suited for CO metabolism, encoding a single bifunctional Ni-CODH within a complete WLP, five additional monofunctional Ni-CODHs, in addition to two monofunctional CODHs on a novel mobile genetic element. The experiments were designed to assess metabolic activity, and no conclusions regarding biomass increase or growth yields are drawn in this study. We demonstrate that anaerobic methanotrophic archaea can function as acetogens and methanogens in the presence of CO with high metabolic activity.

While the inhibitory effect of CO on AOM was previously suggested, earlier studies remained inconclusive due to low reaction rates and the confounding presence of acetogenic microbial community members[48]. Our study not only confirms the suppressive effect of CO but also reveals its capacity to redirect redox flux toward acetogenesis and methanogenesis, reshaping our understanding of ANME metabolic flexibility. The high enrichment of anaerobic methanotrophs in our system, coupled with high CO and $CH_4$ turnover rates and the absence of known methanogens, enabled direct assessment of the impact of CO on AOM and ANME physiology.

Alternative CO-oxidizing reactions seemed to be minor contributors to overall CO oxidation. Aerobic CODHs (CoxL) and bifunctional Ni-dependent CODHs were found to be encoded in several metagenome-assembled genomes from our culture and were expressed at levels substantially lower than *cooS* and *cdh* in '*Ca*. M. carboxydivorans'; none of these bacteria are known as CO-oxidizers or CO-dependent acetate or formate producers. Highly expressed *cooS* genes and unique *mcr* presence reinforce the central role of '*Ca*. M. carboxydivorans' in $CH_4$ and CO turnover. CO oxidation appears tightly linked to the expression of five monofunctional Ni-CODHs, while a sixth CODH, part of the bifunctional CODH/ACS complex, is likely involved in acetogenic metabolism. Two chromosomal CODHs (CooS_3 and CooS_5) were hardly expressed, suggesting either functional redundancy or differential regulation under conditions not tested. The low

amino acid identity of these two CODHs compared to the other CODHs and their different structural group within CODHs (Supplementary Data 10) might indicate different substrate affinity. As CO can be toxic, both by imposing redox pressure and through interference with metal-containing enzymes, it is plausible that a subset of monofunctional CODHs functions as a high-Km/high-Vmax system as proposed for methanogens[36]. This would oxidize CO at elevated concentrations until non-inhibitory concentrations for AOM. Whether CODHs in *Ca*. M. carboxydivorans' primarily or selectively support detoxification or energy conservation remains unresolved and will require targeted physiological studies.

The redox pressure induced by CO oxidation has important implications for central energy metabolism in ANME. Oxidation of CO to $CO_2$ generates a high flux of reduced ferredoxin, a low-potential electron carrier, and if not redistributed at an equivalent rate, this could lead to metabolic arrest[37]. Under $CO/NO_3^-$ conditions reduced redox carriers were re-oxidized in the nitrate-dependent electron transport chain; the electron flow from ferredoxin into the membrane might happen through $F_{420}H_2$ and the Fqo system, the cytochrome $bc_1$ complex, and the canonical or non-canonical nitrate reductases, as well as contribute to energy conservation through the establishment of a $H^+$ motive force (Fig. 5). Under $CO/CO_2$ conditions the redox pressure must be relieved through alternative, non-respiratory, pathways. The observed rapid formate production provides a short-term non-toxic sink for excess electrons without energy conservation. Acetate formation is required for re-oxidizing redox carriers and conserving ATP, demonstrating acetogenic metabolism in anaerobic methanotrophic archaea coupled to energy conservation. Methane production by anaerobic methanotrophic archaea identified in our study extends beyond the previously suggested metabolic backflux, or methane production with net methane oxidation[33]. Our metabolite profile resembles that of *Methanosarcina acetivorans* during growth on CO[31], illustrating that the *Methanoperedenaceae* methanotrophic

Mcr is functionally fully reversible despite the extensive post-translational modifications compared to the known methanogenic Mcr enzymes[20]. Functional studies should assess whether the *Methanoperedenaceae* Mcr exhibits different kinetic properties compared to its methanogenic counterparts; however, we clearly demonstrate that it is highly active in the methanogenic direction, with no lag time after switching from methanotrophy.

A complementary study on CO metabolism in a syntrophic consortium with marine ANME[49] showed CO oxidation by the syntrophic marine *Methanocomedenaceae* (ANME-2b), including incorporation of CO-derived carbon into biomass as assessed by FISH-nanoSIMS. CO oxidation rates decreased at higher partial pressures of CO, indicating some degree of toxicity as no growth was observed under CO conditions, suggesting increased maintenance costs whether it is redox balancing or detoxification. The product profile on CO coupled to $CO_2$ reduction did not yield detectable acetate or formate. Indicating that CO can be directly oxidized and assimilated without obligatory reductive branching to acetate or formate. Together with observations in this study and from *Methanosarcina* methanogens[31,50], these divergent product profiles suggest that downstream routing of reducing equivalents and the balance between growth, maintenance and toxicity are lineage- and condition-dependent. The observed downregulation under $CO/CO_2$ conditions of the key methanogenic energy conservation machinery in this study (e.g. *mcr*, *mtr*) was also identified in *Methanocomedenaceae*[49] and *M. acetivorans* when growing on CO compared to methanol[50]. This leaves open the question on whether CO-dependent methanogenesis by methanotrophs could also lead to net growth and whether this is the case at environmental CO concentrations.

The widespread prevalence of CODHs across *Methanoperedenaceae* suggests CO oxidation is conserved and functionally relevant in freshwater ANME. The phylogenetic analysis of CODHs indicates that multiple marine *Methanocomedenaceae* ANME encode close homologs to the highly expressed CODHs. We hypothesize that marine ANME CO metabolism is similar to the observed CO metabolism in this study, thereby excluding the need for a syntrophic partner when cultivated with CO and $CO_2$. Most, but not all, freshwater anaerobic methanotrophs seem to encode the genetic potential to couple CO oxidation to nitrate respiration, either via a canonical or a non-canonical nitrate reductase. Hydrogen is not an obligatory intermediate as '*Ca*. M. carboxydivorans' lacks genes encoding functional hydrogenases[35]. This is consistent with the lack of hydrogen production in *Deferribacter autotrophicus* during CO oxidation coupled to nitrate reduction[26] but unlike acetogenic metabolism in many bacteria[51]. Our findings extend nitrate respiratory CO oxidation capability to the archaeal domain and suggest that redox partitioning drives the versatile metabolism of ANME under non-respiratory conditions.

The discovery of multiple CODHs and redox-associated enzymes on a circular MGE in '*Ca*. M. carboxydivorans' suggests that core metabolic modules can be maintained and expressed extrachromosomally. All catabolism-related genes on the MGE, including additional CODHs, nitrate and nitrite reductases, and Hdr−Fdh complexes, are encoded within a near-identical region to the host genome, suggesting recent duplication, mobilization, or lateral acquisition. If shared across *Methanoperedenaceae*, such an MGE could significantly augment central metabolism. A similar phenomenon was observed in *Thermoanaerobacter kivui*, where a mobile megatransposon conferred a complete and highly expressed CO oxidation module, directly enhancing CO metabolism and growth[52]. The MGE in this study could, in a similar fashion, promote CO metabolism, allowing flexible gene regulation and preserving essential redox functions under variable environmental conditions, as was indirectly tested in this study. No evidence for autonomous replication or essential host metabolic functions were identified on the MGE, suggesting host-dependent maintenance. Resolving the acquisition and role of MGEs in relation to their host and metabolism requires complete (circularized) genomes, as duplicated elements are otherwise indistinguishable from the host genome[53]. For most ANME archaea, such data are not yet available and given the potential high similarity to host sequences, this would require long-read sequencing or targeted PCR validation to disentangle. While copy number cannot be directly inferred from our data, the near-identical read coverage suggests a 1:1 ratio with the chromosome. Although this MGE differs in structure, it mirrors the content of core metabolic genes of Borgs which are recently discovered large linear extrachromosomal elements, retrieved from natural environments, which likewise encode extensive redox and energy-conserving machinery[38,41,54]. These seem to expand host metabolic capacity, as supported by lateral gene transfer studies of ANME[40]. The selective environment of a bioreactor may stabilize such elements and offers a promising platform to study functionality. Likely MGEs reflect a broader evolutionary strategy for enhancing metabolic flexibility under fluctuating environmental conditions.

CO is a common but often overlooked compound in anoxic environments, which can be formed by thermal or microbial degradation of organic matter[55–57]. Although CO concentrations are typically low (ppb range), depending on the environment, microbial activity can generate elevated levels (ppm range) in microniches[57,58]. CO is not routinely measured in biogeochemical studies of ANME-inhabited environments, and as a result, its concentrations and spatiotemporal dynamics remain unknown. The prevalence of multiple CODHs across the majority of *Methanoperedenaceae*, including those encoded on an MGE, indicates CO as a recurrent substrate. The inhibitory effect of CO could explain the absence of methane oxidation activity in settings where it would otherwise be expected based on geochemical conditions. CO can be metabolized by a range of microorganisms including aerobic CO oxidizers, ammonia- and methane-oxidizers, as well as anaerobes such as acetogens, methanogens, and sulfate reducers[55,56,59]. The capacity of ANME to oxidize CO adds an unrecognized dynamic to global $CH_4$ and trace gas cycling, potentially extending beyond Earth and supporting hypotheses of Martian methanotrophy[60,61]. These findings highlight the need to re-examine metagenomic and biogeochemical data from ANME-dominated environments and to investigate how CO availability and utilization influence anoxic carbon cycling in both freshwater and marine environments. Further studies are required to resolve the ecological and metabolic consequences of CO for ANME physiology, particularly its potential to interfere with AOM in the anaerobic microbial methane filter.

## Methods

### Inoculum and bioreactor operation

The inoculum used in this study was granular biomass from a nitrate-driven anaerobic methane oxidation reactor enriched with '*Ca*. Methanoperedens BLZ2'[8]. The reactor operates in a sequencing fed-batch mode with a working volume of 8-11 L, maintained at 30 °C, 200 rpm, and a pH of $7.3 \pm 0.1$, controlled by the addition of 1 M sodium carbonate ($Na_2CO_3$). The reactor was continuously sparged with 15 mL min⁻¹ $CH_4/CO_2$ (95:5) and fed with medium (flow rate 2–2.5 L day⁻¹). Once a day, stirring is stopped to allow biomass settling, and after 5 min, excess liquid is removed until a final volume of 8 L is reached. During this study, the reactor was operated with 18 mM nitrate in the medium, corresponding to a loading rate of 4.5-5.6 mM nitrate $L^{-1}$ day$^{-1}$. Under these conditions, nitrite was undetectable. The mineral medium consisted of the following components [g $L^{-1}$]: $MgSO_4$, 0.16; $CaCl_2$, 0.24; and $KH_2PO_4$, 0.05. Trace elements and vitamins were supplied via stock solutions. The trace element stock solution (1000x) contained [g $L^{-1}$]: $FeCl_2 \cdot 4 H_2O$, 1.35; $MnCl_2 \cdot 4 H_2O$, 0.1; $CoCl_2 \cdot 6 H_2O$, 0.024; $CaCl_2 \cdot 2 H_2O$, 0.1; $ZnCl_2$, 0.1; $CuCl_2 \cdot 2 H_2O$, 0.025; $H_3BO_3$, 0.01; $Na_2MoO_4 \cdot 2 H_2O$, 0.024; $NiCl_2 \cdot 6 H_2O$, 0.22; $Na_2SeO_3$, 0.017; $Na_2WO_4 \cdot 2 H_2O$, 0.004; and nitrilotriacetic acid, 12.8.

The vitamin stock solution (10000x) consisted of [mg L$^{-1}$]: biotin, 20; folic acid, 20; pyridoxine-HCl, 100; thiamin-HCl · 2 H$_2$O, 50; riboflavin, 50; nicotinic acid, 50; D-Ca-pantothenate, 50; vitamin B$_{12}$, 2; p-aminobenzoic acid, 50; and lipoic acid, 50. The bioreactor medium supply was continuously sparged with Ar:CO$_2$ in a 95:5 ratio.

## Activity assays with different electron donors

Activity assays were performed using 120 mL serum bottles containing mineral medium supplemented with 20 mM 4-(2-hydroxyethyl)-1-piperazineethanesulfonic acid (HEPES) buffer (pH 7.25). As inoculum, 60 mL aliquots of granular biomass were withdrawn anoxically from the bioreactor. Aliquots were transferred to an anaerobic chamber, washed three times in anoxic medium through settling, followed by liquid removal through decanting[62]. The washed biomass was resuspended in a final volume of 40 mL and sealed with butyl rubber stoppers (Terumo, Leuven, Belgium), the headspace flushed with Ar/CO$_2$ (95:5) for 15 min, and the pressure set to 1.5 bar. Finally, gaseous substrates (CH$_4$, CO) were added to the headspace, with equal partial pressures across conditions for every experiment. The pH was adjusted to 7.25 ± 0.1 with anoxic KOH. Microcosms were incubated at 30 °C and 200 rpm (New Brunswick Innova 40, Eppendorf, Germany). Sodium nitrate (3 mM) was used as the electron acceptor. To prevent nitrite accumulation, nitrate and nitrite concentrations were monitored after four hours and subsequently multiple times per day. Nitrate was added to maintain a working concentration between 0.5 and 3 mM.

Negative controls included medium without biomass to account for gas loss during sampling (biological replicates, $n = 2$) and heat-killed biomass to account for abiotic interactions (biological replicates, $n = 3$). Positive controls without antibiotics consisted of NO$_3^-$ with 20 mL CH$_4$, 20 mL CO and 20 mL CH$_4$/CO 1:1 (biological replicates, $n = 2$). For the comparative electron donor experiment in the presence of nitrate (biological replicates, $n = 4$), 20mL of CH4 (>99%) or 10mL of CO (>99%) with 10mL of Ar/CO2 to account for the difference in partial pressure was added. For the CO/CO$_2$ condition (biological replicates, $n = 3$), nitrate was omitted. To suppress bacterial activity an antimicrobial compound mixture consisting of streptomycin, vancomycin, ampicillin, and kanamycin (SVAK)[62], each prepared at 15 mg mL$^{-1}$, was added with a final concentration of 50 µg mL$^{-1}$ per antibiotic. These experiments (biological replicates, $n = 4$) were repeated for comparative transcriptomics and after 16 hours, with confirmed CH$_4$ or CO oxidation, microcosms were sacrificed for RNA extraction. Biomass was harvested in an anaerobic chamber by opening the microcosms, transferring the liquid into a 50 mL Greiner tube (Greiner Bio-One GmbH, Frickenhausen, Germany), and decanting the supernatant. The remaining pellet was snap-frozen in liquid nitrogen, freeze-dried overnight and stored at −70 °C until RNA extraction.

## Analytical methods

For each experiment using granular biomass, the biomass dry weight concentration in the reactor was measured by filtering a 10 mL reactor aliquot (technical replicates, $n = 3$) through pre-dried Whatman glass microfiber filters (Maidstone, UK), followed by drying for 48 hours at 80–100 °C. The CH$_4$ concentration was measured using 50 µL gas-phase samples analyzed on a gas chromatograph equipped with a flame ionization detector (Hewlett-Packard 5890, Palo Alto, CA, USA) and a Porapak Q100 column, with the injector and oven temperatures set to 150 °C and 120 °C, respectively. Samples (technical replicates, $n = 3$) were analyzed with automatic peak integration using the GC ChemStation software (Agilent Technologies, Santa Clara, CA, USA). The CO and CO$_2$ concentrations were measured using 100 µL gas-phase samples analyzed on a gas chromatograph equipped with a thermal conductivity detector (Agilent Technologies 6890, Santa Clara, CA, USA) in a dual-column configuration consisting of a Porapak

Q and Molesieve 13x connected in series. Samples (technical replicates, $n = 1$) were analyzed with automatic peak integration using the GC ChemStation software. Total CH$_4$, CO and CO$_2$ in both liquid- and gas-phase was determined using Henry's law with a Henry constant ($H^{cp}$) for CH$_4$, CO and CO$_2$ of $1.45 \times 10^{-5}$, $9.09 \times 10^{-6}$ and $2.97 \times 10^{-4}$ mol m$^{-3}$ Pa$^{-1}$, respectively[63]. The consumption rates were calculated by linear least-squares regression and normalized with biomass dry weight. For CO oxidation rates, shorter time intervals were selected depending on the observed oxidation rates. Rates were corrected for gas loss through sampling using a negative control with only medium and both CH$_4$ and CO (10.5 and 32.6 µmol d$^{-1}$ gDW$^{-1}$, respectively). Negative controls, including double autoclaved biomass, showed no decrease in CO concentrations (uncorrected: 21.87 µmol d$^{-1}$ gDW$^{-1}$, so equal to sampling effect), excluding abiotic interactions. For comparison, methane oxidation rates observed in this study were in the same order of magnitude as CH$_4$ oxidation rates observed in previous work using the same ANME enrichment[62]. Nitrate and nitrite in the liquid-phase were monitored (technical replicates, $n = 1$) using MQuant colorimetric test strips (Merck, Darmstadt, Germany) with a lower detection limit of 10 and 2 mg L$^{-1}$, for nitrate and nitrite, respectively. Volatile fatty acids were quantified from liquid samples using high-performance liquid chromatography on a Shimadzu LC2030C Plus system (Shimadzu, Kyoto, Japan) equipped with a differential refractive index detector (RID-20A) and a Shodex SH1821 column operated at 45 °C. The eluent was 5 mM sulfuric acid at a flow rate of 0.8 mL min$^{-1}$.

## Nucleic acid extraction and sequencing

DNA was extracted from granular bioreactor biomass sampled at the same time as the inoculum used for the activity assays. Prior to extraction, granules were manually ground to disrupt the biomass, after which DNA was extracted using the DNeasy PowerSoil Kit (Qiagen, Hilden, Germany), and the eluted DNA was stored at −20 °C. Freeze-dried biomass from the microcosm cultivations was manually ground to disrupt the granules, followed by extraction using the RNeasy PowerSoil Kit (Qiagen, Hilden, Germany). RNA samples were treated with DNAase I at 37 °C for 30 min according to the RiboPure Bacteria Kit protocol (Thermo Fisher Scientific, Waltham, MA, USA) and stored at −70 °C. DNA and RNA quality were assessed using a NanoDrop Spectrophotometer ND-1000 (Isogen Life Science, Utrecht, The Netherlands) and a Bioanalyzer 2100 (Agilent, Santa Clara, CA, United States), respectively. Nucleic acid concentrations were quantified using a Qubit 2.0 fluorometer with the dsDNA HS assay for DNA and the RNA HS assay for RNA (Thermo Fisher Scientific, Waltham, MA, United States).

DNA and RNA sequencing were performed by Macrogen Europe BV (Amsterdam, The Netherlands) using the NovaSeq X platform. For whole-genome sequencing, the library was prepared with the PCR-free TruSeq kit (550 bp insert) using the provided DNA. For RNA sequencing, the library was prepared using the TruSeq Stranded kit with the NEB rRNA Depletion Kit (bacteria). Long-read sequencing was performed using the MinION Mk1C at Radboud University's in-house facility. Libraries were prepared using DNA extracted from earlier obtained reactor biomass with the Ligation Sequencing Kit 1D (SQK-LSK109) in combination with the Native Barcoding Expansion Kit (EXP-NBD104).

## Metagenome analysis

**Quality control and assembly of obtained reads.** Nanopore long-read sets were quality filtered using Filtlong v0.2.1 (https://github.com/rrwick/Filtlong) with the –min_length 1000 and –keep_percent 90 settings. Flye v2.8.3[64] was used to assemble the filtered long read dataset using the –meta setting (metaFlye). In addition, Aviary v0.9.0 (https://github.com/rhysnewell/aviary) was also used for hybrid assembly and binning of short- and long-reads, internally calling several different assemblers and binning tools.

**Binning and polishing of obtained MAGs, including 'Ca. M. BLZ2' and a mobile genetic element.** Metagenome-assembled genome (MAG) contiguity was visualized using Bandage v0.8.1[65]. In the assembly graph, the 'Ca. M. BLZ2's contig (3.68 Mbp) and a putative extrachromosomal contig (156-kbp) shared a region (129-kbp) that was double the coverage. This suggests two scenarios: two separate circular sequences, or the 'Ca. M. BLZ2's contig contains two identical 129-kbp regions. Based on mapping alignment data, long reads mapped to the 'Ca. M. BLZ2's contig and the shared region were extracted and reassembled with metaFlye v2.8.3, resulting in a circular contig of 3.80-Mbp. Likewise, long reads mapped to the putative mobile genetic element (MGE) and shared region were reassembled into another circular contig of 156-kbp. These contigs were further polished with a single round of racon v1.4.21[66] and three rounds of pilon5 v1.2.4[67]. Four other MAGs were circularized in the Flye assembly and polished with a single round of Racon v1.4.21 and three rounds with Pilon5 v1.2.4. Obtained closed and polished genomes, including the putative MGE, were merged with the recovered genomes from the hybrid assembly. These were dereplicated with the 'dereplicate' workflow in dRep v3.4.2[68] with completeness >40% and default settings. This non-redundant genome set (Supplementary Data 1) consisted of five circularized genomes and 31 MAGs, of which 21 were high-quality (>90% complete and <5% contaminated), 14 were medium-quality (>50% complete and <10% contaminated), and one was low-quality (>40% complete and <10% contaminated), as estimated by CheckM2[69]. Remapping of trimmed Illumina reads on this non-redundant genome set using CoverM v0.7.0[70] showed 91.4% of the reads mapped (Supplementary Data 2). Phylogenetic classification of the bacterial and archaeal genomes was assigned using GTDB-Tk v2.4.0 (release R220) using the classify_wf command. Genomes were annotated using Prokka v1.14.6[71] and EggNOG v5.0 with eggNOG-mapper v2.1.12[72,73]. Functional annotation of the bacterial side-community was done with KofamScan v1.3.0[74] against the KEGG Orthology HMM database using an e-value threshold of 1e-5. KO annotations of the bacterial community linked to microbial metabolism of interest are listed in Supplementary Data 3. Circular genomes were explored and visualized using Proksee[75], GC skew was calculated in Proksee using GC Skew v1.0.2 with window-size 1000 and step-size 10 settings.

**Primer design and junction-PCR to confirm mobile genetic element.** To confirm the circularity and validate the transitions between unique and shared regions between the chromosome and the MGE, primers were manually designed to amplify the two predicted junctions of the shared 129-kbp region on the MGE. Primer pairs targeted the left junction (LJ-F/R, positions 64149-64843) and right junction (RJ-F/R positions 91243–91892). Primers were ordered from Biolegio (Nijmegen, The Netherlands) and characteristics are listed in Supplementary Table 1. DNA previously extracted from the enrichment culture was used as template. Thermal cycling included an initial denaturation at 96 °C for 5 min, followed by 35 cycles of denaturation at 96 °C for 40 s, annealing at 60.9 °C for 30 s, and elongation at 72 °C for 30 s, with a final elongation at 72 °C for 5 min. Amplicon sizes were verified by gel electrophoresis on a 1.5% agarose gel run at 80 V for 30 min, showing a single band at the expected size for each primer set. Bands were excised and purified using the GeneJET Gel Extraction Kit (Thermo Fisher Scientific, Waltham, MA, United States). Purified amplicons were sequenced by Sanger sequencing (BaseClear, Leiden, The Netherlands), and the resulting sequences were aligned with the sequenced MGE, confirming junction transitions.

**Comparative and structural analysis of the mobile genetic element.** The genome relatedness between the here presented 'Ca. M. BLZ2' genome and identified mobile genetic element were compared to earlier obtained mobile genetic elements and the recently polished genome of 'Ca. M. nitroreducens'[7,38,41]. This was done by a protein-based alignment using high-scoring segment pairs established by using DiGAlign v2.0[76]. The resulting dendrogram was visualized in iTOL v6.9.1[77], mid-point rooted, and branch length ignored for visualization purposes. Viral signatures within identified MGE were identified using Phigaro v2.4.0 with default parameters[78]. Detection of imperfect tandem repeats was performed using ETANDEM with default parameters from the EMBOSS suite v5.0.0[79].

**Identification and structural classification of carbon monoxide dehydrogenases (CODH).** Assembled MAGs were searched for both aerobic (molybdenum-dependent) Mo-CODH and anaerobic (nickel-dependent) Ni-CODH protein sequences. As query sequence for Mo-CODH, the active site subunit CoxL from *Oligotropha carboxidovorans* (WP_013913730.1) was selected[80]. Only CoxL sequences containing the active-site AYXCSFR (form I) or AYXGAGR (form II) motifs[59] were assigned to reported CoxL forms. For Ni-CODH protein sequences the same steps were taken as described by Inoue et al.[81]. *Carboxydothermus hydrogenoformans* CooSII (WP_011343033.1) was used as a query for the CooS-type CODH and the *Methanosarcina barkeri* Acetyl-CoA synthase α subunit as a query for the CdhA-type CODH (WP_011305243.1). Identified Ni-CODH sequences in this study were merged with the 1942 non-redundant Ni-CODH protein sequences from Inoue et al. 2019 and aligned using MAFFT v7.525[82] with the E-INS-I method. The alignment was trimmed using trimAl v1.5.0[83] with the gap threshold 0.9 setting. Identified sequences without deletions in the D-, B-, and C-clusters and catalytic residues were classified as Ni-CODHs, with structural classification based on variation patterns[81]. To verify whether the identified Ni-CODHs were part of a CODH/ACS gene cluster, HMM profiles for *acsABCDE* and *cdhABCDE* were used to scan for co-localized cluster genes[23]. To investigate the prevalence of Ni-CODH, the analysis for active site preservation was repeated for all publicly available genomes of GTDB species representatives within *Methanoperedenaceae* from GTDB release R220, following reannotation with Prokka v1.14.6[71].

**Phylogenetic classification of CODHs of 'Ca. M. BLZ2'.** Identified Ni-CODH sequences of 'Ca. M. BLZ2' were merged with the non-redundant Ni-CODH protein sequences from Inoue et al. 2019. Protein sequences derived from metagenome-assembled genomes (MAGs) that did not pass quality checks in the Genome Taxonomy Database (GTDB)[84] were removed. The remaining sequences were dereplicated using Diamond v2.1.13[85] cluster at 95% identity and 95% coverage. Ni-CODH sequences identified in 'Ca. M. BLZ2' were then searched against the GTDB R220 proteome dataset to recover closely related homologs, as well as against the earlier obtained mobile genetic elements[38,41]. Homologs annotated as K00192 or K00198 by KofamScan v1.3.0[74] were retained. All sequences were aligned using Muscle v5.0.1278[86] and subsequently trimmed with trimAl v1.5.0[83] with the gap threshold 0.9 setting. A phylogenetic tree was inferred with VeryFastTree v4.0.5[87] under the WAG substitution model, employing the approximate maximum-likelihood method. Node support was evaluated using 1000 local bootstrap resamples. The resulting tree was visualized using iTOL v6.9.1[77].

**Metatranscriptome analysis.** Raw sequence data was analyzed using TranscriptM v0.4.0 (https://github.com/sternp/transcriptm), a comprehensive suite for trimming, filtering, rRNA removal and mapping, using the obtained non-redundant genome set as input. Relative abundance was calculated using CoverM v0.7.0. Count matrices with mapped reads were obtained using FeatureCounts v2.0.1[88] with paired-end reads enabled. Further analysis, including data visualization, was done in RStudio v4.4.0. Mapped reads were converted into transcripts per million (TPM) to compare expression levels for the complete datasets and per genome of interest. Differential gene expression analysis was done using the R package DESeq2 1.44.0[89,90]. Differentially

expressed genes were identified as genes with an expression change of >1.5-fold and an adjusted *P*-value ≤0.05. Gene cluster visualization was done using the R package gggenes and heatmaps were generated with the R package pheatmap and for visualization scaled per row. For additional visualization in heatmaps the TPM values were averaged per condition.

## Reporting summary

Further information on research design is available in the Nature Portfolio Reporting Summary linked to this article.

## Data availability

The metagenomic and metatranscriptomic data have been deposited in the European Nucleotide Archive database under study accession https://www.ebi.ac.uk/ena/browser/view/PRJEB97457, with identifiers: SAMEA120155624, SAMEA120631852 and SAMEA120155615–120155623. The metagenome-assembled genomes (MAGs) generated in this study are under the same study accession, with identifiers SAMEA121075817-SAMEA121075852. The assembled mobile genetic element is part of SAMEA121075817. The full non-redundant set of MAGs including associated metadata, has also been deposited on Figshare (https://doi.org/10.6084/m9.figshare.30903899). Source data are provided with this paper.

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

## Acknowledgements

This work and both R.A.E. and C.U.W. were supported by the SIAM Gravitation Grant (024.002.002) and a VIDI Talent Grant (VI.Vidi.223.012). S.J.M., G.W.T., and A.O.L. were supported by the Australian Research Council (ARC) Future Fellowship (FT190100211), Laureate Fellowship (FL170100086), and DECRA fellowship (DE250101094), respectively. We thank Diana Sousa and Tom Schonewille (Wageningen University & Research) for supplying carbon monoxide. We acknowledge the Queensland University of Technology for funding and maintaining the high-performance computing cluster used for bioinformatic analyses. We are grateful to Martijn Diender and Anastasia Galani for insightful discussions on CODHs, Aytan Rustamova for support during her MSc internship, and Aharon Oren for advice on prokaryotic nomenclature. We also thank Peter ter Horst, Martijn Wissink, and Conall Holohan for bioreactor maintenance during visits to the Centre for Microbiome Research at the Queensland University of Technology.

## Author contributions

R.A.E. and C.U.W. designed the experiments and wrote the manuscript. R.A.E. performed the laboratory work and led data interpretation. H.L., A.O.L., and R.A.E. performed the bioinformatic analysis and contributed to data analysis and interpretation. G.W.T. and S.J.M. hosted and supported the bioinformatic work and contributed to data interpretation. All authors discussed the results, contributed to writing of the manuscript, and approved the final manuscript.

## Competing interests

The authors declare no competing interests.
