## [Transparent Peer Review file · Nature Communications]

Carbon monoxide metabolism in freshwater anaerobic methanotrophic archaea

Corresponding Author: Dr Cornelia Welte

Version 0:

Reviewer comments:

Reviewer #1

(Remarks to the Author)

The manuscript reports the enrichment of ANME-2d and identifies a new physiological activity: CO oxidation. Considering the important role of ANME in methane removal and its implications for climate change, this finding is interesting, as it may represent a potential alternative survival strategy for ANME-2d in the environment. From the perspective of Methanosarcinales physiology (the broader clade including ANME-2 and methanogens), CO utilization has been studied for many years. CO can serve as both a substrate and a toxin for methanogens: while it can be metabolized, high concentrations of CO typically inhibit growth, and biomass yields are often limited during growth on CO. This characteristic was not reflected in the present study, where the authors observed high CO depletion activity by ANME-2d both in the presence and absence of an electron acceptor.

In the presence of nitrate, CO was rapidly depleted, and genes (*cooS*) involved in CO metabolism were highly expressed. This provides strong support for active CO oxidation by ANME-2d. However, *cooS* has previously been described as a gene involved in CO detoxification (DOI: 10.1007/s00203-007-0266-1). It remains unclear whether ANME-2d cells actually grew in the CO-amended enrichment. Do nitrate and CO support growth, or merely detoxification activity? Are there increases in cell numbers or gene copy numbers? Are there any stress-response genes or pathways highly expressed? Additional evidence is needed to clarify the role of *cooS* in CO metabolism by ANME-2d.

In the absence of nitrate, CO depletion was still rapid, and fermentation products and methane were produced, similar to the behavior of *Methanosarcina acetivorans*. Because both gene sets of *cooS* and *cdh/acs* are present in ANME-2d, the specific roles and relationship between these genes in CO transformation remain unclear. Do ANME-2d cells require *cooS* when nitrate is absent, given that *cooS* was not highly expressed under CO/CO₂ compared to CH₄/NO₃⁻ conditions when CO was absent? Can *cdh* directly convert environmental CO to acetyl-CoA without first reducing CO₂ to CO, as occurs in *M. acetivorans*? Furthermore, do ANME-2d cells grow in enrichments with CO/CO₂, or have they evolved the ability to survive without electron acceptors? These questions should be discussed in more detail.

In the incubations with CO/CO₂ and CH₄/NO₃⁻, an unexpected observation was that expression of most genes related to nitrate metabolism (Table S14) was lower or comparable in CH₄/NO₃⁻ incubations than in CO/CO₂ incubations, despite the absence of nitrate in the CO/CO₂ setup. Additionally, the metatranscriptomic samples were taken at 16 h, yet it seems that methane consumption had not begun by that time (Fig. 1c). Is there any link for these phenomenon? It seems that the physiological states of the incubations are different.

The authors also report that CO inhibits AOM, but the underlying mechanism is not explained. Are there any supporting data are presented to substantiate this inhibition? And a more detailed discussion is strongly recommended.

More specific comments

Line 16–18: Misleading, methane oxidation by ANME-2d in the absence of nitrate or presence of nitrate?.

Line 180: Where were the MAGs obtained, from the CO enrichment or from bioreactors? Are these MAGs related to high CO concentrations?

Line 274/276: How is 20% were calculated? You have 80 ml headspace with 1.5 bar, then added 20 ml CO, the vol percentage should be ~14%.

Line 305: Is the plasmid host-dependent or obligately associated with the host? Does the plasmid encode essential genes

absent from the host genome?

Fig. 5: Are there any pyruvate lyase genes that could support formate production?

Reviewer #2

(Remarks to the Author)

This is an excellent study that uncovers a previously unknown physiological adaptation and metabolic pathway in anaerobic methane oxidising archaea (ANME), i.e., the ability of *Candidatus Methanoperedens* BLZ2 to oxidise CO. Previously, it had been suggested that CO may inhibit CH₄ oxidation by ANME archaea. Here, the authors show that BLZ2 can oxidise CO and that BLZ2 even preferentially oxidises CO in cultures fed CH₄ and CO simultaneously.

The findings represent a major advance in identifying metabolic flexibility in these organisms and expanding the fundamental knowledge of the environmental niche some ANME occupy. Not only do the experiments demonstrate that CO is oxidised both under nitrate reducing as well as nitrate depleted conditions, the BLZ2 dominated culture oxidised CO at rates up to 20x higher than methane in the presence of nitrate. With CO being a well-known degradation product of biomass under anoxic conditions, this is a significant finding because the presence of these organisms in environments where both methane and CO are available as substrates simultaneously, either in the presence or absence of nitrate, the environmental significance of CO utilisation by *M'peredens* requires us to reevaluate our concepts of methane and CO cycling in anoxic habitats.

The characterisation of CO metabolism by *M'peredens* is based on a multipronged experimental approach using highly enriched bioreactor culture of *M'peredens* and (i) measuring CH₄ and CO oxidation in the presence of nitrate as well as in the absence of nitrate and presence of CO₂. The contribution of bacterial members of the consortium (*M'peredens* is not available as a pure culture) to the CO metabolism is addressed through inhibition with a mix of different antibiotics, which demonstrates that CO oxidation rates are actually higher when commensal bacteria are inhibited. The work reveals that in the absence of nitrate as respiratory electron acceptor and presence of CO₂, *M'peredens* produces formate, acetate and methane as products of CO metabolism.

Metagenomic sequencing of the culture resulted in a closed genome of BLZ2 and the identification of an extrachromosomal mobile genetic element; these are present in a 1:1 ratio and encode several several anaerobic Ni-CO-dehydrogenases, both as components of the CODH/acetyl Co A synthetase complex as well as the standalone CooS type. Other key enzymes detected include those of the complete Wood-Ljungdahl pathway (WLP), and nitrate, nitrite and heterodisulfide reductases are also encoded. Although a number of metagenome assembled genomes (MAGs) are shown to harbour Mo-dependent CODH enzymes as well as finding some CODH/ACS enzyme in *Chloroflexi*, transcript numbers are relatively low suggesting the main CO oxidation is carried by BLZ2 which is shown to express several CooS as well as nitrate reductases. Critically, metatranscriptomic analyses demonstrate expression of the aforementioned enzymes from BLZ2 during cultivation conditions with CO and CO₂ allowing the authors to propose a metabolic model of acetogenic CO metabolism in BLZ2, with additional branches that lead to formation of formate and methane as additional products.

Finally, the authors assess the distribution of Ni-CODH (*cooS* and *cdh* genes) in *M'peredenaceae*, demonstrating these as widely distributed in available genome sequences, suggesting CO metabolism to be a widespread feature of these methanotrophs, a significant fraction also having *nar* genes encoding nitrate reductases. Furthermore, their phylogenetic analyses of CODH enzymes shows their presence in some marine ANME lineages too which may indicate the relevance of this metabolic flexibility extending to marine ANME.

Overall, the work as well as the manuscript are excellent.
Minor comments are outlined below.

Line – comment

43-46 reorder argument for clarity? Recently, formate was proposed as an alternative donor coupled to nitrate reduction¹⁶. Intriguingly, methane oxidation was completely halted by formate addition. However, the enrichment also contained '*Ca. Methylospirillum oxyfera*' and '*Ca. Kuenenia stuttgartiensis*', both of which encode formate dehydrogenases, with '*Ca. K. stuttgartiensis*' being a potent formate oxidizer¹⁷.

114/115 – clarify whether antibiotic mix was 50ug/mL each

118 – no centrifugation?

224 – branch

232/233 – reword? *Carboxydotherrmus hydrogenoformans* *CooSII* (WP_011343033.1) was used as query for the CooS-type CODH

Figure 1 legend: 1D. how much CO was added after 20 hours?

309 – S5 has carbon balances, should this be S12 (although that's expression)?

327 – could you add reference for the Borgs here (for those of us who are not yet familiar with these)

366-368 – did you assess whether these were form 1 and/or form 2?

Fig 4 legend - could you add hypothetical protein for Hyp in Figure legend; define genes that have not been mentioned elsewhere (*hao*, ...?)

Fig 6 – for ease of reference, could show BLZ2 in bold in tree?

Point-by-point response to reviewer comments

Our point-by-point responses are provided in blue.

We thank both reviewers for their thorough and constructive evaluations of our manuscript and for recognizing the significance of this study in advancing the understanding of metabolic flexibility in anaerobic methanotrophic archaea. We appreciate the time and expertise invested in reviewing our work.

Reviewer 1 raised important points regarding the physiological interpretation of CO metabolism in '*Ca. Methanoperedens carboxydivorans*', particularly in relation to growth, detoxification and the role of mono- or bifunctional CODHs. We have addressed these concerns by expanding and clarifying the relevant sections and by strengthening the contextual explanation of CO-driven redox metabolism in ANME.

Reviewer 2 provided highly supportive feedback and we both appreciate and share the excitement. This feedback gave several helpful clarifications, which have improved the precision and readability of the manuscript.

For both reviewers we have carefully considered each point raised and provide detailed, point-by-point responses below. Corresponding revisions have been incorporated into the main text (colour blue), figures, and supplementary materials where appropriate. These changes have substantially strengthened the manuscript.

REVIEWER COMMENTS

Reviewer #1 (Remarks to the Author):

The manuscript reports the enrichment of ANME-2d and identifies a new physiological activity: CO oxidation. Given both the ecological significance and the ANME in methane removal and its implications for climate change, this finding is interesting, as it may represent a potential alternative survival strategy for ANME-2d in the environment. From the perspective of Methanosarcinales physiology (the broader clade including ANME-2 and methanogens), CO utilization has been studied for many years. CO can serve as both a substrate and a toxin for methanogens: while it can be metabolized, high concentrations of CO typically inhibit growth, and biomass yields are often limited during growth on CO. This characteristic was not reflected in the present study, where the authors observed high CO depletion activity by ANME-2d both in the presence and absence of an electron acceptor.

We appreciate the recognition on the ecological significance of methane oxidation by ANME for mitigating climate change. Given the relative importance and novelty of the field, the discovery of additional metabolic capabilities in these organisms warrants careful investigation. We also acknowledge that questions regarding growth, biomass yields, and toxicity are central to seminal studies of CO metabolism in methanogens and should be done for methanotrophs.

At the same time, we emphasize that ANME physiology differs fundamentally from that of methanogenic Methanosarcinales, both in metabolism and experimental accessibility. Several practical and conceptual constraints are therefore important to clarify. First, ANME are not available as axenic cultures, and the experiments were performed using an enrichment of granular biomass, which inherently limits direct quantification of growth or biomass yields. Second, the aim of the present study was not to assess growth on CO, but to establish whether CO can be actively metabolized by ANME-2d. Within this scope, we provide multiple independent lines of evidence supporting a metabolic role for CO in ANME-2d, including rapid CO depletion, coordinated expression of multiple *cooS* genes under CO-oxidizing conditions, their genomic localization within redox related gene clusters (including a mobile genetic element), increased transcriptomic activity and the stoichiometric conversion of CO into methane, acetate, and formate. Together, these observations do not exclude toxicity or detoxification but seemingly cannot be limited to sole detoxification excluding growth. Combined with the widespread presence of multiple CODH gene copies across Methanoperedenaceae, these findings support CO oxidation as an integrated and physiologically relevant metabolic capability, either for growth, detoxification, or both in ANME-2d.

To avoid any ambiguity and improve clarity of the scope of our work, we have revised the manuscript to remove wording that could be interpreted as just implying growth and addressed toxicity more explicitly throughout the point-by-point responses. Specifically, we have adjusted the text and figure captions as follows:

360: Figure 3. Overall transcriptional activity of microbial community members under different conditions

411: Figure 4. Transcriptional response of *cooS* and *cdh* gene clusters in 'Ca. M. BLZ2' under varying conditions.

442: Figure 5. Hypothesized metabolic model of CO metabolism of 'Ca. M. BLZ2'.

Furthermore, the first paragraph of the Discussion is adjusted to reflect this:

495-498: The experiments were designed to assess metabolic activity and no conclusions regarding biomass increase or growth yields are drawn in this study. We demonstrate that anaerobic methanotrophic archaea can function as acetogens and methanogens in the presence of CO with high metabolic activity.

We fully agree with the reviewer that longer-term experiments explicitly designed to quantify growth, detoxification, and metabolic fluxes will be essential future work. We view the present study as a foundational step that establishes the physiological capability and genetic repertoire for ANME CO oxidation, providing a roadmap for future investigations into ANME physiology, including those directions highlighted by the reviewer.

In the presence of nitrate, CO was rapidly depleted, and genes (*cooS*) involved in CO metabolism were highly expressed. This provides strong support for active CO oxidation by ANME-2d. However, *cooS* has previously been described as a gene involved in CO detoxification (DOI: 10.1007/s00203-007-0266-1). It remains unclear whether ANME-2d cells actually grew in the CO-amended enrichment. Do nitrate and CO support growth, or merely detoxification activity? Are there increases in cell numbers or gene copy numbers? Are there any stress-response genes or pathways highly expressed? Additional evidence is needed to clarify the role of *cooS* in CO metabolism by ANME-2d.

We agree that the rapid depletion of CO in the presence of nitrate, together with the high expression of *cooS* genes, provides strong evidence for active CO oxidation. As outlined above, we cannot assess growth. However, in 'Ca. Methanoperedens carboxydvorans', including associated mobile MGE, *cooS* is encoded seven times; with two gene cluster embedded within redox-active gene clusters that include nitrate reductases and heterodisulfide- and formate dehydrogenase complexes (Figure 4; prevalence shown across ANME-2d in Figure 6). Under CO-oxidizing conditions, some *cooS* genes rank among the most highly expressed genes in the genome and show coordinated expression with nitrate reductases. These gene expression profiles are

inconsistent with a transient detoxification response and supportive of a role for *cooS* in energy-conserving CO oxidation coupled to nitrate.

When examining genes related to cell division and stress physiology, including *ftsZ* and DNA/protein-repair associated genes, comparison of the CH₄/NO₃⁻ to CO/NO₃⁻ conditions did not reveal an increased stress-related response. Instead, expression levels remained overall comparable, arguing against a dominant stress response or detoxification metabolism with CO/NO₃⁻.

To more accurately convey our results and interpretations regarding detoxification vs. growth we incorporated a section on stress response in the results and prior to discussing the CO metabolism added the following statements addressing growth/stress, in addition to recommendations for additional work unravelling toxicity mechanisms in ANME-2d in the future.

Find adjusted results on stress related genes:

393-401: CO is a known toxic gas and can inhibit growth in methanogens, where it can trigger detoxification responses facilitated by *cooS*. However, we did not observe a consistent or pronounced pattern change in expression of canonical stress, DNA and protein repair or cell division associated genes under CO/NO₃⁻ and CO/CO₂ conditions (Supplementary Data 9). While a subset of stress-associated genes was upregulated and expression of some cell-division genes was altered, this pattern is more consistent with a moderate physiological adjustment to CO exposure, or the nitrate depleted redox environment rather than a generalized stress response. Instead, transcriptional changes were dominated by redox and central metabolic pathways. Additionally, under CO/CO₂ conditions, the 'Ca. M. BLZ2' *cooS* genes were downregulated but still expressed at similar TPM compared to the conditions with CH₄/NO₃⁻ (Figure 4).

Find adjusted discussion incorporating the detoxification aspect:

518-523: As CO can be toxic, both by imposing redox pressure and through interference with metal-containing enzymes, it is plausible that a subset of monofunctional CODHs functions as a high-K_m/high-V_{max} system as proposed for methanogens⁷⁷. These would oxidize CO at elevated concentrations, until non-inhibitory concentrations for AOM. Whether CODHs in 'Ca. M. carboxydivorans' primarily or selectively support detoxification or energy conservation remains unresolved and will require targeted physiological studies.

533: The observed rapid formate production provides a short-term non-toxic sink for excess electrons without energy conservation.

In the absence of nitrate, CO depletion was still rapid, and fermentation products and methane were produced, similar to the behavior of *Methanosarcina acetivorans*. Because both gene sets of *cooS* and *cdh/acs* are present in ANME-2d, the specific

roles and relationship between these genes in CO transformation remain unclear. Do ANME-2d cells require *cooS* when nitrate is absent, given that *cooS* was not highly expressed under CO/CO₂ compared to CH₄/NO₃⁻ conditions when CO was absent?

This cannot be resolved with the present data. Under CO/CO₂ conditions (no nitrate), we observed formation of methane, acetate, and formate, which is consistent with an active *cdh/acs* and associated redox pathways as proposed in the manuscript. However, all *cooS* and *cdh/acs* were downregulated under CO/CO₂ compared to the nitrate conditions. Hence we cannot resolve whether nitrate-coupled CO oxidation and nitrate-depleted CO metabolism may rely on different or no *cooS*. Given the high rates and downregulation of *cdh/acs* in tested conditions it is suggestive *cdh/acs* supports acetyl-CoA/acetate formation under nitrate-depleted (CO/CO₂) conditions, while *cooS* likely contribute to CO oxidation/detoxification. Whether this reflects detoxification or growth-related metabolism and how it changes at different CO concentrations needs to be tested in dedicated long-term experiments. Gene deletion studies are currently not available for any member of the ANME so it is not possible to test it directly.

Can *cdh* directly convert environmental CO to acetyl-CoA without first reducing CO₂ to CO, as occurs in *M. acetivorans*? Furthermore, do ANME-2d cells grow in enrichments with CO/CO₂, or have they evolved the ability to survive without electron acceptors? These questions should be discussed in more detail.

In our study we cannot assess this as all tested conditions contained CO₂. While this paper was under review, a very complementary paper on CO metabolism in marine ANME was submitted to bioRxiv (<https://doi.org/10.1101/2025.09.21.677609>). We are aware that it is not peer reviewed yet, but the exciting results warrant incorporation in lieu to the reviewer's questions and our discussion. We complemented our discussion with their findings assessing some of the reviewer's questions:

545-559: "A complementary study on CO metabolism in a syntrophic consortium with marine ANME (currently available as preprint)⁷⁹ showed CO oxidation by the syntrophic marine *Methanocomedenaceae* (ANME-2b), including incorporation of CO-derived carbon into biomass as assessed by FISH-nanoSIMS. CO oxidation rates decreased at higher partial pressures of CO, indicating some degree of toxicity as no growth was observed under CO conditions, suggesting increased maintenance costs for redox balancing or detoxification. The product profile on CO coupled to CO₂ reduction did not yield detectable acetate or formate indicating that CO can be directly oxidized and assimilated without obligatory reductive branching to acetate or formate. Together with observations in this study and from *Methanosarcina* methanogens^{31,80}, these divergent product profiles suggest that downstream routing of reducing equivalents and the balance between growth, maintenance and toxicity are lineage- and condition-dependent. The observed downregulation of the key methanogenic energy conservation machinery in this study (e.g. *mcr*, *mtr*) under CO/CO₂ conditions was also identified in *Methanocomedenaceae*⁷⁹ and *M. acetivorans* when growing on

CO compared to methanol⁸⁰. This leaves open the question on whether CO-dependent methanogenesis by methanotrophs could also lead to net growth and whether this is the case at environmental CO concentrations.”

In the incubations with CO/CO₂ and CH₄/NO₃⁻, an unexpected observation was that expression of most genes related to nitrate metabolism (Table S14) was lower or comparable in CH₄/NO₃⁻ incubations than in CO/CO₂ incubations, despite the absence of nitrate in the CO/CO₂ setup.

Supplementary Data 9 shows that nitrate and nitrite reductases (*narGHJI* and *nrfHA*) are downregulated when comparing the CH₄/NO₃⁻ to CO/CO₂ conditions. Good to note is that this is the transcriptome profile which switched profoundly, likely due to the impact of switching from a nitrate-rich environment to a nitrate-depleted environment. Indicating a physiological transition, consistent with this interpretation, the transcriptome under CO/CO₂ conditions differed strongly from both respiratory conditions, with coordinated changes in central carbon metabolism, nitrogen metabolism, and redox homeostasis (lines 393-396). The switch away from nitrate respiration likely altered regulatory states and redox balance, as we suggest in our metabolic model due to increased electron flux through reduced ferredoxin and change in terminal electron acceptor, which may also influence the expression of CODH genes given their genomic co-localization and possible functional coupling with nitrate reductases.

391-393: The transcriptome under CO/CO₂ conditions was profoundly different from the two respiratory conditions with many changes in central carbon metabolism, nitrogen metabolism, and redox homeostasis. Likely caused by switching from a nitrate-rich to nitrate-depleted environment after prolonged incubation on nitrate.

Additionally, the metatranscriptomic samples were taken at 16 h, yet it seems that methane consumption had not begun by that time (Fig. 1c). Is there any link for these phenomenon? It seems that the physiological states of the incubations are different.

The metatranscriptomic samples were taken at 16 h to capture the early physiological and regulatory state close to the start of the batch incubations. The biomass originated from a bioreactor which dependent on nitrate-driven AOM for years. At the 16 h sampling point, methane oxidation was already transcriptionally active in the CH₄/NO₃⁻ condition, as indicated by high expression of key methane-oxidation genes, including *mcr*, despite relatively low initial methane turnover rates. A slight decrease in methane was already detectable at 16 h, and the overall time-course clearly demonstrates methane consumption (Fig. 1c; source data). Sampling at later time points would increasingly reflect secondary effects of prolonged incubation, such as substrate depletion, product accumulation, and physiological adaptation, making cross-condition comparisons less directly interpretable.

The authors also report that CO inhibits AOM, but the underlying mechanism is not explained. Are there any supporting data are presented to substantiate this inhibition?

The inhibition of AOM is supported by the halting of methane oxidation under CO amended conditions and the preferential utilization of CO when both substrates are present. In terms of mechanism, we mention the preference of CO over CH₄ in multiple instances (line 67-68, 281, 490-492). The specific selection of preference over mechanistic inhibition results from our observation of methanogenesis in CO/CO₂ conditions indicating that the core methanotrophic/methanogenic machinery remains functional and is not directly inhibited by CO. The apparent inhibition of AOM is therefore best explained by substrate preference.

To make this clearer we adjusted the initial line of the discussion:

490-492: In this study, we discovered that CO is a potent inhibitor of AOM for 'Ca. Methanoperedens BLZ2', henceforth called 'Ca. M. carboxydivorans', as it is preferentially used when CH₄ and CO are present.

And a more detailed discussion is strongly recommended.

We have revised the discussion at multiple points and believe that the constructive and critical feedback has led to a clearer and more detailed discussion. At the same time, we note that Nature Communications imposes practical constraints on manuscript length. Within these limits, we have aimed to provide a balanced discussion that addresses the key implications of the findings without extending beyond the scope of the study.

Please find a point-by-point response to the more specific comments below:

More specific comments

Line 16–18: Misleading, methane oxidation by ANME-2d in the absence of nitrate or presence of nitrate?.

16-18: Without respiratory nitrate, CO oxidation led to acetogenesis and methanogenesis with rates comparable to methane oxidation with nitrate.

Line 180: Where were the MAGs obtained, from the CO enrichment or from bioreactors? Are these MAGs related to high CO concentrations?

The MAGs, including both archaeal and bacterial members of the community, were reconstructed from DNA extracted from granular bioreactor biomass sampled at the

same time as the inoculum used for the activity assays. These MAGs were therefore not derived from the short-term CO incubation experiments. Consequently, they do not represent adaptation to elevated CO concentrations during batch incubations yet reflect the genomic potential of a long-term nitrate-dependent ANME enrichment culture.

While these MAGs originate from an ANME enrichment, CO concentrations are not routinely measured in ANME-inhabited environments. As a result, it is currently not possible to link either the ANME MAGs or the bacterial MAGs to specific CO concentration niches *in situ*. The relationship of bacterial MAGs being known CO-oxidizers is unknown and discussed in line **500-503**. The limited biogeochemical CO data is discussed in the final paragraph of the discussion.

Adjusted line for clarifying DNA extraction:

152-155: DNA was extracted from granular bioreactor biomass sampled at the same time as the inoculum used for the activity assays. Prior to extraction, granules were manually ground to disrupt the biomass, after which DNA was extracted using the DNeasy PowerSoil Kit (Qiagen, Hilden, Germany), and the eluted DNA was stored at $-20\text{ }^{\circ}\text{C}$.

Line 274/276: How is 20% were calculated? You have 80 ml headspace with 1.5 bar, then added 20 ml CO, the vol percentage should be ~14%.

Thanks for noticing this. The adjusted percentages are:

275: "(14.3% vol/vol in the headspace)"

277: "(7.1% vol/vol each)"

286: "(7.1% vol/vol)"

Line 305: Is the plasmid host-dependent or obligately associated with the host? Does the plasmid encode essential genes absent from the host genome?

The mobile genetic element (MGE) shows similar coverage relative to the *Methanoperedens* chromosome, indicating stable co-occurrence in the bioreactor enrichment. Although a putative MCM helicase was identified within the unique region of the MGE it showed very low transcriptome coverage across conditions and is not accompanied by identifiable replication machinery hence it is not included in the main text. Combined this indicates that the MGE is host-dependent. The MGE does not encode genes that are essential for its host. Instead, it primarily carries accessory redox- and energy-related functions, which are as shown in the manuscript within the shared regions.

To clarify this we adjusted the discussion accordingly:

584-585: No evidence for autonomous replication or essential host metabolic functions were identified on the MGE, suggesting host-dependent maintenance

Fig. 5: Are there any pyruvate lyase genes that could support formate production?

We did not detect pyruvate lyase genes either on the chromosome or on the MGE, instead there are multiple formate dehydrogenase gene clusters and a pyruvate:ferredoxin oxidoreductase as depicted in Figure 5.

Reviewer #2 (Remarks to the Author):

This is an excellent study that uncovers a previously unknown physiological adaptation and metabolic pathway in anaerobic methane oxidising archaea (ANME), i.e., the ability of *Candidatus Methanoperedens BLZ2* to oxidise CO. Previously, it had been suggested that CO may inhibit CH₄ oxidation by ANME archaea. Here, the authors show that BLZ2 can oxidise CO and that BLZ2 even preferentially oxidises CO in cultures fed CH₄ and CO simultaneously.

The findings represent a major advance in identifying metabolic flexibility in these organisms and expanding the fundamental knowledge of the environmental niche some ANME occupy. Not only do the experiments demonstrate that CO is oxidised both under nitrate reducing as well as nitrate depleted conditions, the BLZ2 dominated culture oxidised CO at rates up to 20x higher than methane in the presence of nitrate. With CO being a well-known degradation product of biomass under anoxic conditions, this is a significant finding because the presence of these organisms in environments where both methane and CO are available as substrates simultaneously, either in the presence or absence of nitrate, the environmental significance of CO utilisation by *M'peredens* requires us to reevaluate our concepts of methane and CO cycling in anoxic habitats.

The characterisation of CO metabolism by *M'peredens* is based on a multipronged experimental approach using highly enriched bioreactor culture of *M'peredens* and (i) measuring CH₄ and CO oxidation in the presence of nitrate as well as in the absence of nitrate and presence of CO₂. The contribution of bacterial members of the consortium (*M'peredens* is not available as a pure culture) to the CO metabolism is addressed through inhibition with a mix of different antibiotics, which demonstrates that CO oxidation rates are actually higher when commensal bacteria are inhibited. The work reveals that in the absence of nitrate as respiratory electron acceptor and presence of CO₂, *M'peredens* produces formate, acetate and methane as products of CO metabolism.

Metagenomic sequencing of the culture resulted in a closed genome of BLZ2 and the identification of an extrachromosomal mobile genetic element; these are present in a 1:1 ratio and encode several several anaerobic Ni-CO-dehydrogenases, both as components of the CODH/acetyl CoA synthetase complex as well as the standalone CooS type. Other key enzymes detected include those of the complete Wood-Ljungdahl pathway (WLP), and nitrate, nitrite and heterodisulfide reductases are also encoded. Although a number of metagenome assembled genomes (MAGs) are shown to harbour Mo-dependent CODH enzymes as well as finding some CODH/ACS enzyme in Chloroflexi, transcript numbers are relatively low suggesting the main CO oxidation is carried by BLZ2 which is shown to express several CooS as well as nitrate reductases. Critically, metatranscriptomic analyses demonstrate expression of the aforementioned enzymes from BLZ2 during cultivation conditions with CO and CO₂ allowing the authors to propose a metabolic model of acetogenic CO metabolism in BLZ2, with additional branches that lead to formation of formate and methane as additional products.

Finally, the authors assess the distribution of Ni-CODH (cooS and cdh genes) in M'peredenaceae, demonstrating these as widely distributed in available genome sequences, suggesting CO metabolism to be a widespread feature of these methanotrophs, a significant fraction also having nar genes encoding nitrate reductases. Furthermore, their phylogenetic analyses of CODH enzymes shows their presence in some marine ANME lineages too which may indicate the relevance of this metabolic flexibility extending to marine ANME.

Overall, the work as well as the manuscript are excellent. Minor comments are outlined below.

We appreciate the supportive feedback and share the excitement. Please find our point-by-point responses below, thanks for helping fix some of the ambiguity and clarifying questions.

Line – comment

43-46 reorder argument for clarity? Recently, formate was proposed as an alternative donor coupled to nitrate reduction¹⁶. Intriguingly, methane oxidation was completely halted by formate addition. However, the enrichment also contained 'Ca. Methyloirabilis oxyfera' and 'Ca. Kuenenia stuttgartiensis', both of which encode formate dehydrogenases, with 'Ca. K. stuttgartiensis' being a potent formate oxidizer¹⁷.

43-46: Recently, formate was proposed as an alternative donor coupled to nitrate reduction¹⁶. Intriguingly, methane oxidation was completely halted by formate addition. However, the enrichment also contained 'Ca. Methyloirabilis oxyfera' and 'Ca. Kuenenia stuttgartiensis', both of which encode formate dehydrogenases, with 'Ca. K. stuttgartiensis' being a potent formate oxidizer¹⁷.

114/115 – clarify whether antibiotic mix was 50ug/mL each

115-116: To suppress bacterial activity an antimicrobial compound mixture consisting of streptomycin, vancomycin, ampicillin, and kanamycin (SVAK)³³, each prepared at 15 mg mL⁻¹, was added with a final concentration of 50 µg mL⁻¹ per antibiotic.

118 – no centrifugation?

Indeed, no centrifugation was used; due to the granular structure of the biomass, rapid settling occurs by gravity alone. This approach also minimized handling time, avoided temperature increases associated with aliquoting into multiple microcentrifuge tubes, which could affect RNA integrity and transcriptional state. Using wide-diameter Greiner tubes enabled efficient anoxic settling and decanting without noticeable biomass loss (see also lines 97-100), allowing biomass to be harvested, halting metabolic activity by transfer to ice-cold medium and the pellet to be snap-frozen in liquid nitrogen within approximately 5 minutes.

224 – branch

224-225: “The resulting dendrogram was visualized in iTOL v6.9.¹⁵⁰, mid-point rooted and branch length ignored for visualization purposes.”

232/233 – reword? *Carboxydotherrmus hydrogenoformans* CooSII (WP_011343033.1) was used as query for the CooS-type CODH

236-238: *Carboxydotherrmus hydrogenoformans* CooSII (WP_011343033.1) was used as query for the CooS-type CODH and the *Methanosarcina barkeri* Acetyl-CoA synthase α subunit as query for the CdhA-type CODH (WP_011305243.1).

Figure 1 legend: 1D. how much CO was added after 20 hours?

302-305: Figure legend “D, CO with CO₂ as electron acceptor and 50 µg mL⁻¹ antibiotics (SVAK) cocktail (biological replicates, n =3); increase in CO₂ is depicted, after 20 hours an additional 5 mL CO was added. Error bars represent standard deviation.”

This is also depicted in the Supplementary Data 4 ‘Correction for CO refeed (5 mL 100% CO at 20h)’.

309 – S5 has carbon balances, should this be S12 (although that’s expression)? Apologies for caused inconvenience, that should have been Supplementary Figure 1 where the multiple sequence alignment of obtained CODH alignments is depicted.

309-310: “The genome contained five gene clusters encoding Ni-dependent carbon monoxide dehydrogenase (CODH) subunits (Supplementary Figure 1).”

327 – could you add reference for the Borgs here (for those of us who are not yet familiar with these)

References are added per request, good to note these are also presented in Supplementary Figure 3.

329-331: “Distinct from other ANME-associated mobile genetic elements (MGEs), including Borgs (Supplementary Fig. 3)^{47,48}, this MGE encodes a unique combination of respiratory enzymes relevant for CO oxidation and nitrate reduction.”

366-368 – did you assess whether these were form 1 and/or form 2?

Thank for this suggestion. We did not assess whether they were form 1 or form 2 given their low transcript abundance. We have redone this and distinguished the putative *coxL* genes by the presence of an AYXCSFR active-site motif to allocate form I, and an AYXGAGR motif for form II. In total 15/25 initial putative Mo-CODHs had a conserved form I/II motif. We adjusted the following to accommodate this change:

233-235: Only CoxL sequences containing the active-site AYXCSFR (form I) or AYXGAGR (form II) motifs⁵⁴ were assigned to reported CoxL forms.

Supplementary Data 6 now contains an additional column depicting whether the Mo-CODH is form I/II.

This additional cut-off showed only 15 Mo-CODHs (3 Form I, 12 Form II) instead of 25. Furthermore, this reduced the number of Mo-CODH encoding MAGs to 11 instead of 13. Given the low transcript abundance, the exact split of Form I/II is not mentioned in the main text, the adjusted number of Mo-CODHs and Mo-CODH encoding MAGs are mentioned and corrected accordingly:

368-370: In total, 15 Mo-CODHs (*coxL*) were identified across 11 MAGs in our study in addition to one non-ANME monofunctional Ni-CODH (*cooS*) (Supplementary Fig. 1 and Supplementary Data 6).

Fig 4 legend - could you add hypothetical protein for Hyp in Figure legend; define genes that have not been mentioned elsewhere (*hao*, ...?)

This has been corrected for Figure 2, where the text introduces partially these. For the genes that were not introduced, the legend of both Figure 2 and Figure 4 is adjusted:

344-347: Figure 2. Left: Circularized mobile genetic element (MGE) associated with 'Ca. M. BLZ2', encoding multiple gene clusters with redox modules, including *cooS*, *narGHJI*, *nrfHA*, and a soluble heterodisulfide reductase cluster adjacent to *fdhAB*. Hyp. denotes genes encoding proteins of unknown function (hypothetical proteins), and hao indicates a hydroxylamine oxidoreductase-like multi-heme cytochrome.

411-416: Figure 4. Transcriptional response of *cooS* and *cdh* gene clusters in 'Ca. M. BLZ2' under varying conditions. Left: Overview of different gene clusters which encode Ni-CODH. The CODH/ACS *cdhABC* cluster is at the top including the chromosomal Ni-CODHs, the bottom CODH and nitrate reductase cluster was identified on the MGE. Hyp. denotes genes encoding proteins of unknown function (hypothetical proteins), Hao a hydroxylamine oxidoreductase-like multi-heme cytochrome, Ck a carbamate kinase and PrmC a putative PrmC-like methyltransferase.

Fig 6 – for ease of reference, could show BLZ2 in bold in tree?

Incorporated the change, BLZ2 is depicted in bold. Find the adjusted legend also below:

484: Bootstrap values represent support percentages from 1000 replicates. For each genome, estimated genome completeness (%) and contig count are indicated, with 'Ca. M. BLZ2' depicted in bold.